# Meta-Learning with
# Self-Improving Momentum Target

**Jihoon Tack**[1], **Jongjin Park**[1], **Hankook Lee**[1], **Jaeho Lee**[2], **Jinwoo Shin**[1]
[1]Korea Advanced Institute of Science and Technology (KAIST)
[2]Pohang University of Science and Technology (POSTECH)
{jihoontack,jongjin.park,hankook.lee,jinwoos}@kaist.ac.kr
jaeho.lee@postech.ac.kr

## Abstract

The idea of using a separately trained target model (or *teacher*) to improve the performance of the student model has been increasingly popular in various machine learning domains, and meta-learning is no exception; a recent discovery shows that utilizing task-wise target models can significantly boost the generalization performance. However, obtaining a target model for each task can be highly expensive, especially when the number of tasks for meta-learning is large. To tackle this issue, we propose a simple yet effective method, coined *Self-improving Momentum Target (SiMT)*. SiMT generates the target model by adapting from the temporal ensemble of the meta-learner, i.e., the momentum network. This momentum network and its task-specific adaptations enjoy a favorable generalization performance, enabling *self-improving* of the meta-learner through knowledge distillation. Moreover, we found that perturbing parameters of the meta-learner, e.g., dropout, further stabilize this self-improving process by preventing fast convergence of the distillation loss during meta-training. Our experimental results demonstrate that SiMT brings a significant performance gain when combined with a wide range of meta-learning methods under various applications, including few-shot regression, few-shot classification, and meta-reinforcement learning. Code is available at https://github.com/jihoontack/SiMT.

## 1 Introduction

Meta-learning [51] is the art of extracting and utilizing the knowledge from the distribution of tasks to better solve a relevant task. This problem is typically approached by training a meta-model that can transfer its knowledge to a task-specific solver, where the performance of the meta-model is evaluated on the basis of how well each solver performs on the corresponding task. To learn such meta-model, one should be able to (a) train an appropriate solver for each task utilizing the knowledge transferred from the meta-model, and (b) accurately evaluate the performance of the solver. A standard way to do this is the so-called $\mathcal{S}/\mathcal{Q}$ (support/query) protocol [55, 34]: for (a), use a set of *support set* samples to train the solver; for (b), use another set of samples, called *query set* samples to evaluate the solver[1].

Recently, however, an alternative paradigm—called $\mathcal{S}/\mathcal{T}$ (support/target) protocol—has received much attention [58, 62, 32]. The approach assumes that the meta-learner has an access to task-specific *target models*, i.e., an expert model for each given task, and uses these models to evaluate task-specific solvers by measuring the discrepancy of the solvers from the target models. Intriguingly, it has been observed that such knowledge distillation procedure [43, 21] helps to improve the meta-generalization performance [62], in a similar way that such teacher-student framework helps to avoid overfitting under non-meta-learning contexts [30, 24].

---

[1]We give an overview of terminologies used in the paper to guide readers new to this field (see Appendix A).

36th Conference on Neural Information Processing Systems (NeurIPS 2022).

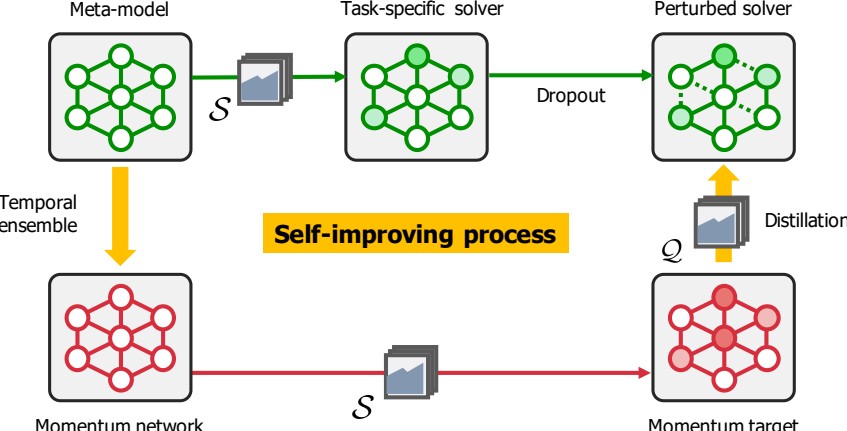

Figure 1: An overview of the proposed *Self-improving Momentum Target (SiMT)*: the momentum network efficiently generates the target model, and by distilling knowledge to the task-specific solver, it forms a self-improving process. $\mathcal{S}$ and $\mathcal{Q}$ denote the support and query datasets, respectively.

Despite such advantage, the $\mathcal{S}/\mathcal{T}$ protocol is difficult to be used in practice, as training target models for each task usually requires an excessive computation, especially when the number of tasks is large. Prior works aim to alleviate this issue by generating target models in a compute-efficient manner. For instance, Lu et al. [32] consider the case where the learner has an access to a model pre-trained on a global data domain that covers most tasks (to be meta-trained upon), and propose to generate task-wise target models by simply fine-tuning the model for each task. However, the method still requires to compute for fine-tuning on a large number of tasks, and more importantly, is hard to be used when there is no effective pre-trained model available, e.g., a globally pre-trained model is usually not available in reinforcement learning, as collecting "global" data is a nontrivial task [9].

In this paper, we ask whether we can generate the task-specific target models by (somewhat ironically) using meta-learning. We draw inspiration from recent observations in semi/self-supervised learning literature [50, 16, 5] that the temporal ensemble of a model, i.e., the momentum network [27], can be an effective teacher of the original model. It turns out that a similar phenomenon happens in the meta-learning scenario: one can construct a momentum network of the meta-model, whose task-specific adaptation is an effective target model from which the task-specific knowledge can be distilled to train the original meta-model.

**Contribution.** We establish a novel framework, coined *Meta-Learning with Self-improving Momentum Target (SiMT)*, which brings the benefit of the $\mathcal{S}/\mathcal{T}$ protocol to the $\mathcal{S}/\mathcal{Q}$-like scenario where task-specific target models are not available (but have access to query data). The overview of SiMT is illustrated in Figure 1. In a nutshell, SiMT is comprised of two (iterative) steps:

- *Momentum target*: We generate the target model by adapting from the momentum network, which shows better adaptation performance than the meta-model itself. In this regard, generating the target model becomes highly efficient, e.g., one single forward is required when obtaining the momentum target for ProtoNet [45].
- *Self-improving process*: The meta-model enables to improve through the knowledge distillation from the momentum target, and this recursively improves the momentum network by the temporal ensemble. Furthermore, we find that perturbing parameters of the task-specific solver of the meta-model, e.g., dropout [47], further stabilizes this self-improving process by preventing fast convergence of the distillation loss during meta-training.

We verify the effectiveness of SiMT under various applications of meta-learning, including few-shot regression, few-shot classification, and meta-reinforcement learning (meta-RL). Overall, our experimental results show that incorporating the proposed method can consistently and significantly improve the baseline meta-learning methods [10, 31, 36, 45]. In particular, our method improves the few-shot classification accuracy of Conv4 [55] trained with MAML [10] on mini-ImageNet [55] from 47.33% → 51.49% for 1-shot, and from 63.27% → 68.74% for 5-shot, respectively. Moreover, we show that our framework could even notably improve on the few-shot regression and meta-RL tasks, which supports that our proposed method is indeed domain-agnostic.

## 2 Related work

**Learning from target models.** Learning from an expert model, i.e., the target model, has shown its effectiveness across various domains [30, 35, 65, 52]. As a follow-up, recent papers demonstrate that meta-learning can also be the case [58, 62]. However, training independent task-specific target models is highly expensive due to the large space of task distribution in meta-learning. To this end, recent work suggests pre-training a global encoder on the whole meta-training set and finetune target models on each task [32]; however, they are limited to specific domains and still require some computations, e.g., they take more than 6.5 GPU hours to pre-train only 10% of target models while ours require 2 GPU hours for the entire meta-learning process (ProtoNet [45] of ResNet-12 [34]) on the same GPU. Another recent relevant work is bootstrapped meta-learning [11], which generates the target model from the meta-model by further updating the parameters of the task-specific solver for some number of steps with the query dataset. While the bootstrapped target models can be obtained efficiently, their approach is specialized in gradient-based meta-learning schemes, e.g., MAML [10]. In this paper, we suggest an efficient and more generic way to generate the target model during the meta-training.

**Learning with momentum networks.** The idea of temporal ensembling, i.e., the momentum network, has become an essential component of the recent semi/self-supervised learning algorithms [3, 5]. For example, Mean Teacher [50] first showed that the momentum network improves the performance of semi-supervised image classification, and recent advanced approaches [2, 46] adopted this idea for achieving state-of-the-art performances. Also, in self-supervised learning methods which enforce invariance to data augmentation, such momentum networks are widely utilized as a target network [19, 16] to prevent collapse by providing smoother changes in the representations. In meta-learning, a concurrent work [6] used stochastic weight averaging [23] (a similar approach to the momentum network) to learn a low-rank representation. In this paper, we empirically demonstrate that the momentum network shows better adaptation performance compare to the original meta-model, which motivates us to utilize it for generating the target model in a compute-efficient manner.

## 3 Problem setup and evaluation protocols

In this section, we formally describe the meta-learning setup under consideration, and $\mathcal{S}/\mathcal{Q}$ and $\mathcal{S}/\mathcal{T}$ protocols studied in prior works.

**Problem setup: Meta-learning.** Let $p(\tau)$ be a distribution of tasks. The goal of meta-learning is to train a meta-model $f_\theta$, parameterized by the meta-model parameter $\theta$, which can transfer its knowledge to help to train a *solver* for a new task. More formally, we consider some *adaptation subroutine* $\texttt{Adapt}(\cdot, \cdot)$ which uses both information transferred from $\theta$ and the task-specific dataset (which we call *support set*) $\mathcal{S}^\tau$ to output a task-specific solver as $\phi^\tau = \texttt{Adapt}(\theta, \mathcal{S}^\tau)$. For example, the model-agnostic meta-learning algorithm (MAML; [10]) uses the adaptation subroutine of taking a fixed number of SGD on $\mathcal{S}^\tau$, starting from the initial parameter $\theta$. In this paper, we aim to give a general meta-learning framework that can be used in conjunction with any adaptation subroutine, instead of designing a method specialized for a specific one.

The objective is to learn a nice meta-model parameter $\theta$ from a set of tasks sampled from $p(\tau)$ (or sometimes the task distribution itself), such that the expected loss of the task-specific adaptations is small, i.e., $\min_\theta \mathbb{E}_{\tau \sim p(\tau)}[\ell^\tau(\texttt{Adapt}(\theta, \mathcal{S}^\tau))]$, where $\ell^\tau(\cdot)$ denotes the test loss on task $\tau$. To train such meta-model, we need a mechanism to evaluate and optimize $\theta$ (e.g., via gradient descent). For this purpose, existing approaches take one of two approaches: the $\mathcal{S}/\mathcal{Q}$ protocol or the $\mathcal{S}/\mathcal{T}$ protocol.

$\mathcal{S}/\mathcal{Q}$ **protocol.** The majority of the existing meta-learning frameworks (e.g., [55, 34]) splits the task-specific training data into two, and use them for different purposes. One is the support set $\mathcal{S}^\tau$ which is used to perform the adaptation subroutine. Another is the *query set* $\mathcal{Q}^\tau$ which is used for evaluating the performance of the adapted parameter and compute the gradient with respect to $\theta$. In other words, given the task datasets $(\mathcal{S}_1, \mathcal{Q}_1), (\mathcal{S}_2, \mathcal{Q}_2), \dots, (\mathcal{S}_N, \mathcal{Q}_N),$[2] the $\mathcal{S}/\mathcal{Q}$ protocol solves

$$\min_\theta \frac{1}{N} \sum_{i=1}^{N} \mathcal{L}\big(\texttt{Adapt}(\theta, \mathcal{S}^{\tau_i}), \mathcal{Q}^{\tau_i}\big), \tag{1}$$

where $\mathcal{L}(\phi, \mathcal{Q})$ denotes the empirical loss of a solver $\phi$ on the dataset $\mathcal{Q}$.

---

[2]Here, while we assumed a static batch of tasks for notational simplicity, the expression is readily extendible to the case of a *stream of tasks* drawn from $p(\tau)$.

$\mathcal{S}/\mathcal{T}$ **protocol.** Another line of work considers the scenario where the meta-learner additionally has an access to a set of *target models* $\phi_{\texttt{target}}$ for each training task [58, 32]. In such case, one can use a teacher-student framework to regularize the adapted solver to behave similarly (or have low *prediction discrepancy*, equivalently) to the target model. Here, a typical practice is to *not split* each task dataset and measure the discrepancy using the support dataset that is used for the adaptation [32]. In other words, given the task datasets $\mathcal{S}_1, \mathcal{S}_2, \ldots, \mathcal{S}_N$ and the corresponding target models $\phi_{\texttt{target}}^{\tau_1}, \phi_{\texttt{target}}^{\tau_2}, \ldots, \phi_{\texttt{target}}^{\tau_N}$, the $\mathcal{S}/\mathcal{T}$ protocol updates the meta-model by solving

$$\min_{\theta} \frac{1}{N} \sum_{i=1}^{N} \mathcal{L}_{\texttt{teach}}\big(\texttt{Adapt}(\theta, \mathcal{S}^{\tau_i}), \phi_{\texttt{target}}^{\tau_i}, \mathcal{S}^{\tau_i}\big), \tag{2}$$

where $\mathcal{L}_{\texttt{teach}}(\phi, \phi_{\texttt{target}}, \mathcal{S})$ denotes a discrepancy measure between the adapted model $\phi$ and the target model $\phi_{\texttt{target}}$, measured using the dataset $\mathcal{S}$.

## 4   Meta-learning with self-improving momentum target

In this section, we develop a compute-efficient framework which bring the benefits of $\mathcal{S}/\mathcal{T}$ protocol to the settings where we do not have access to target-specific tasks or a general pretrained model, as in general $\mathcal{S}/\mathcal{Q}$-like setups. In a nutshell, our framework iteratively generates a *meta-target model* which generalizes well when adapted to the target tasks, by constructing a momentum network [50] of the meta-model itself. The meta-model is then trained, using both the knowledge transferred from the momentum target and the knowledge freshly learned from the query sets. We first briefly describe our meta-model update protocol (Section 4.1), and then the core component, coined *Self-Improving Momentum Target (SiMT)*, which efficiently generates the target model for each task (Section 4.2).

### 4.1   Meta-model update with a $\mathcal{S}/\mathcal{Q}$-$\mathcal{S}/\mathcal{T}$ hybrid loss

To update the meta-model, we use a hybrid loss function of the $\mathcal{S}/\mathcal{Q}$ protocol (1) and the $\mathcal{S}/\mathcal{T}$ protocol (2). Formally, let $(\mathcal{S}_1, \mathcal{Q}_1), (\mathcal{S}_2, \mathcal{Q}_2), \ldots, (\mathcal{S}_N, \mathcal{Q}_N)$ be given task datasets with support-query split, and let $\phi_{\texttt{target}}^{\tau_1}, \phi_{\texttt{target}}^{\tau_2}, \ldots, \phi_{\texttt{target}}^{\tau_N}$ be task-specific target models generated by our target generation procedure (which will be explained with more detail in Section 4.2). We train the meta-model as

$$\min_{\theta} \frac{1}{N} \sum_{i=1}^{N} \bigg( (1-\lambda) \cdot \mathcal{L}(\texttt{Adapt}(\theta, \mathcal{S}^{\tau_i}), \mathcal{Q}^{\tau_i}) + \lambda \cdot \mathcal{L}_{\texttt{teach}}(\texttt{Adapt}(\theta, \mathcal{S}^{\tau_i}), \phi_{\texttt{target}}^{\tau_i}, \mathcal{Q}^{\tau_i}) \bigg), \tag{3}$$

where $\lambda \in [0, 1)$ is the weight hyperparameter. We note two things about Eq. 3. First, while we are training using the target model, we also use a $\mathcal{S}/\mathcal{Q}$ loss term. This is because our method trains the meta-target model and the meta-model simultaneously from scratch, instead of requiring fully-trained target models. Second, unlike in the $\mathcal{S}/\mathcal{T}$ protocol, we evaluate the discrepancy $\mathcal{L}_{\texttt{teach}}$ using the query set $\mathcal{Q}^{\tau_i}$ instead of the support set, to improve the generalization performance of the student model. In particular, the predictions of adapted models on query set samples are softer (i.e., having less confidence) than on support set samples, and such soft predictions are known to be beneficial on the generalization performance of the student model in the knowledge distillation literature [64, 49].

### 4.2   SiMT: Self-improving momentum target

We now describe the algorithm we propose, SiMT (Algorithm 1), to generate the target model in a compute-efficient manner. In a nutshell, SiMT is comprised of two iterative steps: *momentum target* and *self-improving process*. To efficiently generate a target model, SiMT utilizes the temporal ensemble of the network, i.e., the momentum network, then distills the knowledge of the generated target model into the task-specific solver of the meta-model to form a self-improving process.

**Momentum target.** For the compute-efficient generation of target models, we utilize the momentum network $\theta_{\texttt{moment}}$ of the meta-model. Specifically, after every meta-model training iteration, we compute the exponential moving average of the meta-model parameter $\theta$ as

$$\theta_{\texttt{moment}} \leftarrow \eta \cdot \theta_{\texttt{moment}} + (1-\eta) \cdot \theta, \tag{4}$$

where $\eta \in [0, 1)$ is the momentum coefficient. We find that $\theta_{\texttt{moment}}$ can adapt better than the meta-model $\theta$ itself and observe that the loss landscape has flatter minima (see Section 5.5), which can

---

**Algorithm 1** SiMT: Self-Improving Momentum Target

---

**Require:** Distribution over tasks $p(\tau)$, adaptation subroutine $\texttt{Adapt}(\cdot)$, momentum coefficient $\eta$,
   weight hyperparameter $\lambda$, dropout probability $p$, task batch size $N$, learning rate $\beta$.

---

1: Initialize $\theta$ using the standard initialization scheme.
2: Initialize the momentum network with the meta-model parameter, $\theta_{\texttt{moment}} \leftarrow \theta$.
3: **while** not done **do**
4:    Sample $N$ tasks $\{\tau_i\}_{i=1}^N$ from $p(\tau)$
5:    **for** $i = 1$ to $N$ **do**
6:       Sample support set $\mathcal{S}^{\tau_i}$ and query set $\mathcal{Q}^{\tau_i}$ from $\tau_i$
7:       $\phi_{\texttt{moment}}^{\tau_i} = \texttt{Adapt}(\theta_{\texttt{moment}}, \mathcal{S}^{\tau_i})$.                    ▷ Generate a momentum target.
8:       $\phi^{\tau_i} = \texttt{Adapt}(\theta, \mathcal{S}^{\tau_i})$.                    ▷ Adapt a task-specific solver.
9:       $\phi_{\texttt{drop}}^{\tau_i} = \texttt{Dropout}(\phi^{\tau_i}, p)$.                    ▷ Perturb the solver.
10:       $\mathcal{L}_{\texttt{total}}^{\tau_i}(\theta) = (1 - \lambda) \cdot \mathcal{L}(\phi_{\texttt{drop}}^{\tau_i}, \mathcal{Q}^{\tau_i}) + \lambda \cdot \mathcal{L}_{\texttt{teach}}(\phi_{\texttt{drop}}^{\tau_i}, \phi_{\texttt{moment}}^{\tau_i}, \mathcal{Q}^{\tau_i})$   ▷ Compute loss.
11:    **end for**
12:    $\theta \leftarrow \theta - \frac{\beta}{N} \cdot \nabla_\theta \sum_{i=1}^N \mathcal{L}_{\texttt{total}}^{\tau_i}(\theta)$.                    ▷ Train the meta-model.
13:    $\theta_{\texttt{moment}} \leftarrow \eta \cdot \theta_{\texttt{moment}} + (1 - \eta) \cdot \theta$.                    ▷ Update the momentum network.
14: **end while**

---

be a hint for understanding the generalization improvement [29, 12]. Based on this, we propose to generate the task-specific target model, i.e., the momentum target $\phi_{\texttt{moment}}$, by adapting from the momentum network $\theta_{\texttt{moment}}$. For a given support set $\mathcal{S}$, we generate the target model for each task as

$$\phi_{\texttt{moment}}^{\tau_i} = \texttt{Adapt}(\theta_{\texttt{moment}}, \mathcal{S}^{\tau_i}), \qquad \forall i \in \{1, 2, \ldots, N\}. \tag{5}$$

We remark that generating momentum targets does not require an excessive amount of compute (see Section 5.5), e.g., ProtoNet [45] requires a single forward of a support set, and MAML [10] requires few-gradient steps without second-order gradient computation for the adaptation.

**Self-improving process via knowledge distillation.** After generating the momentum target, we utilize its knowledge to improve the generalization performance of the meta-model. To this end, we choose the knowledge distillation scheme [21], which is simple yet effective across various domains, including meta-learning [32]. Here, our key concept is that the momentum target *self-improves* during the training due to the knowledge transfer. To be specific, the knowledge distillation from the momentum target improves the meta-model itself, which recursively improves the momentum through the temporal ensemble. Formally, for a given query set $\mathcal{Q}$, we distill the knowledge of the momentum target $\phi_{\texttt{moment}}$ to the task-specific solver of the meta-model $\phi$ as

$$\mathcal{L}_{\texttt{teach}}(\phi, \phi_{\texttt{moment}}, \mathcal{Q}) := \frac{1}{|\mathcal{Q}|} \sum_{(x,y) \in \mathcal{Q}} l_{\texttt{KD}}\big(f_{\phi_{\texttt{moment}}}(x), f_\phi(x)\big), \tag{6}$$

where $l_{\texttt{KD}}$ is the distillation loss and $|\cdot|$ is the cardinality of the set. For regression tasks, we use the MSE loss, i.e., $l_{\texttt{KD}}(z_1, z_2) := \|z_1 - z_2\|_2^2$, and for classification tasks, we use the KL divergence with temperature scaling [17], i.e., $l_{\texttt{KD}}(z_1, z_2) := T^2 \cdot \texttt{KL}(\sigma(z_1/T) \| \sigma(z_2/T))$, where $T$ is the temperature hyperparameter, $\sigma$ is the softmax function and $z_1, z_2$ are logits of the classifier, respectively. We present the detailed distillation objective of reinforcement learning tasks in Appendix C. Also, note that optimizing the distillation loss only propagates gradients to the meta-model $\theta$, not to the momentum network $\theta_{\texttt{moment}}$, i.e., known as the stop-gradient operator [5, 7].

Furthermore, we find that the distillation loss (6) sometimes converges too fast during the meta-training, which can stop the self-improving process. To prevent this, we suggest perturbing the parameter space of $\phi$. Intuitively, injecting noise to the parameter space of $\phi$ forces an asymmetricity between the momentum target's prediction, hence, preventing $f_\phi$ and $f_{\phi_{\texttt{moment}}}$ from reaching a similar prediction. To this end, we choose the standard dropout regularization [47] due to its simplicity and generality across architectures and also have shown its effectiveness under distillation research [60]: $\phi_{\texttt{drop}} := \texttt{Dropout}(\phi, p)$ where $p$ is the probability of dropping activations. In the end, we use the perturbed task-specific solver $\phi_{\texttt{drop}}$ and the momentum target $\phi_{\texttt{moment}}$ for our evaluation protocol (3).

Table 1: Few-shot regression results on ShapeNet and Pascal datasets. We report the angular error for ShapeNet, and MSE for Pascal. SiMT use the momentum network at meta-test time. Reported results are averaged over three trials, subscripts denote the standard deviation, and bold denotes the best result of each group.

| Method | ShapeNet | | Pascal | |
|---|---|---|---|---|
| | 10-shot | 15-shot | 10-shot | 15-shot |
| MAML [10] | $29.555_{\pm 0.600}$ | $22.286_{\pm 3.369}$ | $2.612_{\pm 0.280}$ | $2.513_{\pm 0.250}$ |
| MAML [10] + SiMT | $\mathbf{18.913_{\pm 2.655}}$ | $\mathbf{16.100_{\pm 1.318}}$ | $\mathbf{1.462_{\pm 0.230}}$ | $\mathbf{1.229_{\pm 0.074}}$ |
| ANIL [36] | $39.915_{\pm 0.665}$ | $38.202_{\pm 1.388}$ | $6.600_{\pm 0.360}$ | $6.517_{\pm 0.420}$ |
| ANIL [36] + SiMT | $\mathbf{37.424_{\pm 0.951}}$ | $\mathbf{29.478_{\pm 0.212}}$ | $\mathbf{5.339_{\pm 0.321}}$ | $\mathbf{5.007_{\pm 0.145}}$ |
| MetaSGD [31] | $17.353_{\pm 1.110}$ | $15.768_{\pm 1.266}$ | $3.532_{\pm 0.381}$ | $2.833_{\pm 0.216}$ |
| MetaSGD [31] + SiMT | $\mathbf{16.121_{\pm 1.322}}$ | $\mathbf{14.377_{\pm 0.358}}$ | $\mathbf{2.300_{\pm 0.871}}$ | $\mathbf{1.879_{\pm 0.134}}$ |

Table 2: Few-shot in-domain adaptation accuracy (%) on 5-way mini- and tiered-ImageNet. SiMT use the momentum network at meta-test time. Reported results are averaged over three trials, subscripts denote the standard deviation, and bold denotes the best result of each group.

| Model | Method | mini-ImageNet | | tiered-ImageNet | |
|---|---|---|---|---|---|
| | | 1-shot | 5-shot | 1-shot | 5-shot |
| Conv4 [55] | MAML [10] | $47.33_{\pm 0.45}$ | $63.27_{\pm 0.14}$ | $50.19_{\pm 0.21}$ | $66.05_{\pm 0.19}$ |
| | MAML [10] + SiMT | $\mathbf{51.49_{\pm 0.18}}$ | $\mathbf{68.74_{\pm 0.12}}$ | $\mathbf{52.51_{\pm 0.21}}$ | $\mathbf{69.58_{\pm 0.11}}$ |
| | ANIL [36] | $47.71_{\pm 0.47}$ | $63.13_{\pm 0.43}$ | $49.57_{\pm 0.04}$ | $66.34_{\pm 0.28}$ |
| | ANIL [36] + SiMT | $\mathbf{50.81_{\pm 0.56}}$ | $\mathbf{67.99_{\pm 0.19}}$ | $\mathbf{51.66_{\pm 0.26}}$ | $\mathbf{68.88_{\pm 0.08}}$ |
| | MetaSGD [31] | $50.66_{\pm 0.18}$ | $65.55_{\pm 0.54}$ | $52.48_{\pm 1.22}$ | $71.06_{\pm 0.20}$ |
| | MetaSGD [31] + SiMT | $\mathbf{51.70_{\pm 0.80}}$ | $\mathbf{69.13_{\pm 1.40}}$ | $\mathbf{52.98_{\pm 0.07}}$ | $\mathbf{71.46_{\pm 0.12}}$ |
| | ProtoNet [45] | $47.97_{\pm 0.29}$ | $65.16_{\pm 0.67}$ | $51.90_{\pm 0.55}$ | $71.51_{\pm 0.25}$ |
| | ProtoNet [45] + SiMT | $\mathbf{51.25_{\pm 0.55}}$ | $\mathbf{68.71_{\pm 0.35}}$ | $\mathbf{53.25_{\pm 0.27}}$ | $\mathbf{72.69_{\pm 0.27}}$ |
| ResNet-12 [34] | MAML [10] | $52.66_{\pm 0.60}$ | $68.69_{\pm 0.33}$ | $57.32_{\pm 0.59}$ | $73.78_{\pm 0.27}$ |
| | MAML [10] + SiMT | $\mathbf{56.28_{\pm 0.63}}$ | $\mathbf{72.01_{\pm 0.26}}$ | $\mathbf{59.72_{\pm 0.22}}$ | $\mathbf{74.40_{\pm 0.90}}$ |
| | ANIL [36] | $51.80_{\pm 0.59}$ | $68.38_{\pm 0.20}$ | $57.52_{\pm 0.68}$ | $73.50_{\pm 0.35}$ |
| | ANIL [36] + SiMT | $\mathbf{54.44_{\pm 0.27}}$ | $\mathbf{69.98_{\pm 0.66}}$ | $\mathbf{58.18_{\pm 0.31}}$ | $\mathbf{75.59_{\pm 0.50}}$ |
| | MetaSGD [31] | $54.95_{\pm 0.11}$ | $70.65_{\pm 0.43}$ | $58.97_{\pm 0.89}$ | $76.37_{\pm 0.11}$ |
| | MetaSGD [31] + SiMT | $\mathbf{55.72_{\pm 0.96}}$ | $\mathbf{74.01_{\pm 0.79}}$ | $\mathbf{61.03_{\pm 0.05}}$ | $\mathbf{78.04_{\pm 0.48}}$ |
| | ProtoNet [45] | $52.84_{\pm 0.21}$ | $68.35_{\pm 0.29}$ | $61.16_{\pm 0.17}$ | $79.94_{\pm 0.20}$ |
| | ProtoNet [45] + SiMT | $\mathbf{55.84_{\pm 0.57}}$ | $\mathbf{72.45_{\pm 0.32}}$ | $\mathbf{62.01_{\pm 0.42}}$ | $\mathbf{81.82_{\pm 0.12}}$ |

## 5 Experiments

In this section, we experimentally validate the effectiveness of the proposed SiMT by measuring its performance on various meta-learning applications, including few-shot regression (Section 5.1), few-shot classification (Section 5.2), and meta-reinforcement learning (meta-RL; Section 5.3).

**Common setup.** By following the prior works, we chose the checkpoints and the hyperparameters on the meta-validation set for the few-shot learning tasks [33, 56]. For RL, we chose it based on the best average return during the training [10]. We find that the hyperparameters, e.g., momentum coefficient $\eta$ or the weight hyperparameter $\lambda$, are not sensitive across datasets and architectures but can vary on the type of the meta-learning scheme or tasks. We provide further details in Appendix D. Moreover, we report the adaptation performance of the *momentum network* for SiMT.

### 5.1 Few-shot regression

For regression tasks, we demonstrate our experiments on ShapeNet [13] and Pascal [63] datasets, where they aim to predict the object pose of a gray-scale image relative to the canonical orientation. To this end, we use the following empirical loss $\mathcal{L}$ to train the meta-model: the angular loss for the ShapeNet ($\sum_{(x,y)\in\mathcal{Q}}\|\cos(f_\phi(x)) - \cos(y)\|^2 + \|\sin(f_\phi(x)) - \sin(y)\|^2$ ) and the MSE loss for the

Table 3: Few-shot cross-domain adaptation accuracy (%) on ResNet-12 trained under 5-way mini- and tiered-ImageNet. We consider CUB and Cars as cross-domain datasets. SiMT use the momentum network at meta-test time. Reported results are averaged over three trials, subscripts denote the standard deviation, and bold denotes the best result of each group.

| Problem | Method | mini-ImageNet $\rightarrow$ | | tiered-ImageNet $\rightarrow$ | |
|---|---|---|---|---|---|
| | | CUB | Cars | CUB | Cars |
| 1-shot | MAML [10] | $39.50_{\pm0.91}$ | $32.87_{\pm0.20}$ | $42.32_{\pm0.69}$ | $36.62_{\pm0.12}$ |
| | MAML [10] + SiMT | $\mathbf{42.32}_{\pm\mathbf{0.62}}$ | $\mathbf{33.73}_{\pm\mathbf{0.63}}$ | $\mathbf{44.33}_{\pm\mathbf{0.43}}$ | $\mathbf{37.21}_{\pm\mathbf{0.35}}$ |
| | ANIL [36] | $37.30_{\pm0.89}$ | $31.28_{\pm1.03}$ | $42.29_{\pm0.33}$ | $36.27_{\pm0.58}$ |
| | ANIL [36] + SiMT | $\mathbf{38.86}_{\pm\mathbf{0.98}}$ | $\mathbf{32.34}_{\pm\mathbf{0.95}}$ | $\mathbf{44.53}_{\pm\mathbf{1.21}}$ | $\mathbf{36.92}_{\pm\mathbf{0.56}}$ |
| | MetaSGD [31] | $41.98_{\pm0.18}$ | $\mathbf{34.52}_{\pm\mathbf{0.56}}$ | $46.48_{\pm2.10}$ | $38.09_{\pm1.21}$ |
| | MetaSGD [31] + SiMT | $\mathbf{43.50}_{\pm\mathbf{0.89}}$ | $33.92_{\pm0.30}$ | $\mathbf{46.62}_{\pm\mathbf{0.41}}$ | $\mathbf{38.69}_{\pm\mathbf{0.26}}$ |
| | ProtoNet [45] | $41.22_{\pm0.81}$ | $32.79_{\pm0.61}$ | $47.75_{\pm0.56}$ | $37.59_{\pm0.80}$ |
| | ProtoNet [45] + SiMT | $\mathbf{44.13}_{\pm\mathbf{0.30}}$ | $\mathbf{34.53}_{\pm\mathbf{0.40}}$ | $\mathbf{48.89}_{\pm\mathbf{0.65}}$ | $\mathbf{38.07}_{\pm\mathbf{0.42}}$ |
| 5-shot | MAML [10] | $56.17_{\pm0.92}$ | $44.56_{\pm0.79}$ | $65.00_{\pm0.89}$ | $51.08_{\pm0.28}$ |
| | MAML [10] + SiMT | $\mathbf{59.22}_{\pm\mathbf{0.39}}$ | $\mathbf{46.59}_{\pm\mathbf{0.21}}$ | $\mathbf{67.58}_{\pm\mathbf{0.61}}$ | $\mathbf{51.88}_{\pm\mathbf{0.52}}$ |
| | ANIL [36] | $53.42_{\pm0.97}$ | $41.65_{\pm0.67}$ | $62.48_{\pm0.85}$ | $50.50_{\pm1.18}$ |
| | ANIL [36] + SiMT | $\mathbf{56.03}_{\pm\mathbf{1.40}}$ | $\mathbf{45.88}_{\pm\mathbf{0.82}}$ | $\mathbf{66.30}_{\pm\mathbf{0.99}}$ | $\mathbf{54.60}_{\pm\mathbf{0.91}}$ |
| | MetaSGD [31] | $58.90_{\pm1.30}$ | $47.44_{\pm1.55}$ | $70.38_{\pm0.27}$ | $56.28_{\pm0.07}$ |
| | MetaSGD [31] + SiMT | $\mathbf{65.07}_{\pm\mathbf{1.89}}$ | $\mathbf{49.86}_{\pm\mathbf{0.84}}$ | $\mathbf{73.93}_{\pm\mathbf{0.42}}$ | $\mathbf{57.97}_{\pm\mathbf{1.34}}$ |
| | ProtoNet [45] | $57.87_{\pm0.77}$ | $48.06_{\pm1.10}$ | $74.35_{\pm0.93}$ | $57.23_{\pm0.25}$ |
| | ProtoNet [45] + SiMT | $\mathbf{63.85}_{\pm\mathbf{0.76}}$ | $\mathbf{51.67}_{\pm\mathbf{0.29}}$ | $\mathbf{75.97}_{\pm\mathbf{0.09}}$ | $\mathbf{59.01}_{\pm\mathbf{0.50}}$ |

Pascal ($\sum_{(x,y)\in\mathcal{Q}}\|f_\phi(x) - y\|^2$), following the prior works [63, 13]. For the backbone meta-learning schemes we use gradient-based approaches, including MAML [10], ANIL [36], and MetaSGD [31]. For all methods, we train the convolutional neural network with 7 layers [63] and apply dropout regularization [47] before the max-pooling layer for SiMT. Table 1 summarizes the results, showing that SiMT significantly improves the overall meta-learning schemes in all tested cases.

## 5.2 Few-shot classification

For few-shot classification tasks, we use the cross-entropy loss for the empirical loss term $\mathcal{L}$ to train the meta-model $\theta$., i.e., $\sum_{(x,y)\in\mathcal{Q}} l_{\texttt{ce}}(f_\phi(x), y)$ where $l_{\texttt{ce}}$ is the cross-entropy loss. We train the meta-model on mini-ImageNet [55] and tiered-ImageNet [38] datasets, following the prior works [32, 56]. Here, we consider the following gradient-based and metric-based meta-learning approaches as our backbone algorithm to show the wide usability of our method: MAML, ANIL, MetaSGD, and ProtoNet [45]. We train each method on Conv4 [55] and ResNet-12 [34], and apply dropout before the max-pooling layer for SiMT. For the training details, we mainly follow the setups from each backbone algorithm paper. See Appendix D.1 for more details.

**In-domain adaptation.** In this setup, we evaluate the adaptation performance on different classes of the same dataset used in meta-training. As shown in Table 2, incorporating SiMT into existing meta-learning methods consistently and significantly improves the in-domain adaptation performance. In particular, SiMT achieves higher accuracy gains on the mini-ImageNet dataset, e.g., 5-shot performance improves from 63.27% $\rightarrow$ 68.74% on Conv4. We find that this is due to the overfitting issue of backbone algorithms on the mini-ImageNet dataset, where SiMT is more robust to such issues. For instance, when training mini-ImageNet 5-shot classification on Conv4, MAML starts to overfit after the first 40% of the training process, while SiMT does not overfit during the training.

**Cross-domain adaptation.** We also consider the cross-domain adaptation scenarios. Here, we adapt the meta-model on different datasets from the meta-training: we use CUB [57] and Cars [26] datasets. Such tasks are known to be challenging, as there exists a large distribution shift between training and testing domains [18]. Table 3 shows the results. Somewhat interestingly, SiMT also improves the cross-domain adaptation performance of the base meta-learning methods across the considered datasets. These results indicate that SiMT successfully learns the ability to generalize to unseen tasks even for the distributions that highly differ from the training.

Table 4: Comparison with bootstrapped targets (Bootstrap) [16] on few-shot in-domain adaptation tasks. We report the adaptation accuracy (%) of Conv4 trained under 5-way mini- and tiered-ImageNet. SiMT use the momentum network at meta-test time. Reported results are averaged over three trials, subscripts denote the standard deviation, and bold indicates the best result of each group.

| | mini-ImageNet | | tiered-ImageNet | |
| --- | --- | --- | --- | --- |
| Method | 1-shot | 5-shot | 1-shot | 5-shot |
| MAML [10] | $47.33_{\pm 0.45}$ | $63.27_{\pm 0.14}$ | $50.19_{\pm 0.21}$ | $66.05_{\pm 0.19}$ |
| MAML [10] + Bootstrap [16] | $48.68_{\pm 0.33}$ | $68.45_{\pm 0.40}$ | $49.34_{\pm 0.26}$ | $68.84_{\pm 0.37}$ |
| MAML [10] + SiMT | $\mathbf{51.49_{\pm 0.18}}$ | $\mathbf{68.74_{\pm 0.12}}$ | $\mathbf{52.51_{\pm 0.21}}$ | $\mathbf{69.58_{\pm 0.11}}$ |
| ANIL [36] | $47.71_{\pm 0.47}$ | $63.13_{\pm 0.43}$ | $49.57_{\pm 0.04}$ | $66.34_{\pm 0.28}$ |
| ANIL [36] + Bootstrap [16] | $47.74_{\pm 0.44}$ | $65.16_{\pm 0.04}$ | $48.85_{\pm 0.34}$ | $66.09_{\pm 0.07}$ |
| ANIL [36] + SiMT | $\mathbf{50.81_{\pm 0.56}}$ | $\mathbf{67.99_{\pm 0.19}}$ | $\mathbf{51.66_{\pm 0.26}}$ | $\mathbf{68.88_{\pm 0.08}}$ |

## 5.3 Reinforcement learning

The goal of meta-RL is training an agent to quickly adapt a policy to maximize the expected return for unseen tasks using only a limited number of sample trajectories. Since the expected return is usually not differentiable, we use policy gradient methods to update the policy. Specifically, we use vanilla policy gradient [59], and trust-region policy optimization (TRPO; [39]) for the task-specific solver and meta-model, respectively, following MAML [10]. The overall training objective of meta-RL is in Appendix C, including the empirical loss $\mathcal{L}$, and the knowledge distillation loss $\mathcal{L}_{\texttt{teach}}$. We evaluate SiMT on continuous control tasks based on OpenAI Gym [4] environments. In these experiments, we choose MAML as our backbone algorithm, and train a multi-layer perceptron policy network with two hidden layers of size 100 by following the prior setup [10]. We find that the distillation loss is already quite effective even without the dropout regularization, and applying it does not improve more. We conjecture that dropout on such a small network may not be effective as it is designed to reduce the overfitting of large networks [47]. We provide more experimental details in Appendix D.1.

**2D Navigation.** We first evaluate SiMT on a 2D Navigation task, where a point agent moves to different goal positions which are randomly chosen within a 2D unit square. Figure 2 shows the adaptation performance of learned models with up to three gradient steps. These results demonstrate that SiMT could consistently improve the adaptation performance of MAML. Also, SiMT makes faster performance improvements than vanilla MAML with additional gradient steps.

**Locomotion.** To further demonstrate the effectiveness of our method, we also study high-

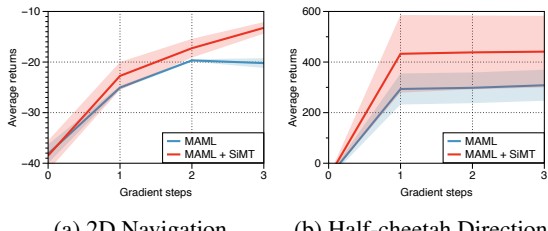

(a) 2D Navigation     (b) Half-cheetah Direction

Figure 2: Meta-RL results for (a) 2D navigation and (b) Half-cheetah locomotion tasks. The solid line and shaded regions represent the truncated mean and standard deviation, respectively, across five runs.

dimensional, complex locomotion tasks based on the MuJoCo [53] simulator. We choose a set of goal direction tasks with a planar cheetah ("Half-cheetah"), following previous works [10, 36]. In the goal direction tasks, the reward is the magnitude of the velocity in either the forward or backward direction, randomly chosen for each task. Figure 2b shows that SiMT significantly improves the adaptation performance of MAML even with a single gradient step.

## 5.4 Comparison with other target models

In this section, we compare SiMT with other meta-learning schemes that utilize the target model, including bootstrapped [16] and task-wise pre-trained target models [32].

**Bootstrapped target model.** We compare SiMT with a recent relevant work, bootstrapped meta-learning (Bootstrap) [16]. The key difference is on how to construct the target model $\phi_{\texttt{target}}$: SiMT utilizes the momentum network while Bootstrap generates the target by *further updating* the parameters of the task-specific solver. Namely, Bootstrap relies on gradient-based adaptation while SiMT does not. This advantage allows SiMT to incorporate various non-gradient-based meta-learning approaches, e.g., ProtoNet [45], as shown in Table 2. Furthermore, SiMT shows not only wider

Table 5: Comparison with pre-trained target models on few-shot in-domain adaptation tasks. We report the adaptation accuracy (%) of ResNet-12 trained on 5-way mini- and tiered-ImageNet. SiMT use the learned momentum network at meta-test time. Reported results are averaged over three trials, subscripts denote the standard deviation, and bold indicates the best result of each group. * indicates the values from the reference and percentage after the bar denotes the proportion of tasks with pre-trained target models for meta-training [32].

| Method | 1-shot train cost (GPU hours) | mini-ImageNet | | tiered-ImageNet | |
|---|---|---|---|---|---|
| | | 1-shot | 5-shot | 1-shot | 5-shot |
| MAML [10]* | 1.31 | $58.84_{\pm 0.25}$ | $74.62_{\pm 0.38}$ | $63.02_{\pm 0.30}$ | $67.26_{\pm 0.32}$ |
| MAML [10] + Lu et al. [32] - 5%* | 5.04 | $59.14_{\pm 0.33}$ | $75.77_{\pm 0.29}$ | $64.52_{\pm 0.30}$ | $68.39_{\pm 0.34}$ |
| MAML [10] + Lu et al. [32] - 10%* | 8.32 | $60.06_{\pm 0.35}$ | $76.34_{\pm 0.42}$ | $\mathbf{65.23}_{\pm 0.45}$ | $70.02_{\pm 0.33}$ |
| MAML [10] + SiMT | 1.64 | $\mathbf{62.05}_{\pm 0.39}$ | $\mathbf{78.77}_{\pm 0.45}$ | $63.91_{\pm 0.32}$ | $\mathbf{77.43}_{\pm 0.47}$ |

Table 6: Ablation study on each component of SiMT. We report the few-shot in-domain adaptation accuracy (%) on Conv4 trained with mini-ImageNet. Here, we use the learned momentum network at meta-test time, except for the first experiment of the table. The reported results are averaged over three trials, subscripts denote the standard deviation, and bold denotes the best result.

| Momentum | Distillation | Dropout | 1-shot | 5-shot |
|---|---|---|---|---|
| - | - | - | $47.33_{\pm 0.45}$ | $63.27_{\pm 0.14}$ |
| ✓ | - | - | $48.98_{\pm 0.32}$ | $66.12_{\pm 0.21}$ |
| ✓ | ✓ | - | $49.23_{\pm 0.24}$ | $66.52_{\pm 0.15}$ |
| ✓ | - | ✓ | $49.25_{\pm 0.41}$ | $65.25_{\pm 0.15}$ |
| ✓ | ✓ | ✓ | $\mathbf{51.49}_{\pm \mathbf{0.18}}$ | $\mathbf{68.74}_{\pm \mathbf{0.12}}$ |

applicability but also better performance than Bootstrap. As shown in Table 4, SiMT consistently outperforms Bootstrap in few-show learning experiments, which implies that the momentum target is more effective than the bootstrapped target.

**Pre-trained target model.** We compare SiMT with Lu et al. [32] which utilizes the task-wise pre-trained target model for meta-learning. To this end, we train SiMT upon a ResNet-12 backbone pre-trained on the meta-training set, by following [32]. As shown in Table 5, SiMT consistently improves over MAML, and more intriguingly, SiMT performs even better than Lu et al. [32]. We conjecture this is because SiMT can fully utilize the target model for all tasks due to its efficiency, while Lu et al. [32] should partially sample the tasks with the target model due to the computational burden: note that when generating target models, SiMT only requires additional 0.3 GPU hours for all tasks while Lu et al. [32] requires more than 3.7 GPU hours for 5% of tasks.

### 5.5 Ablation study

Throughout this section, unless otherwise specified, we perform the experiments in 5-shot in-domain adaptation on mini-ImageNet with Conv4, where MAML is the backbone meta-learning scheme.

**Component analysis.** We perform an analysis on each component of our method in both 1-shot and 5-shot classification on mini-ImageNet: namely, the use of (a) the momentum network $\theta_{\mathtt{moment}}$, (b) the distillation loss $\mathcal{L}_{\mathtt{teach}}$ (6), and (c) the dropout regularization $\mathtt{Dropout}(\cdot)$, by comparing the accuracies. The results in Table 6 show each component is indeed important for the improvement. We find that a naïve combination of the distillation loss and the momentum network does not show significant improvements. But, by additionally applying the dropout, the distillation loss becomes more effective and further improves the performance. Note that this improvement does not fully come from the dropout itself, as only using dropout slightly degrades the performance in some cases.

**Computational efficiency.** Our method may be seemingly compute-inefficient when incorporating meta-learning methods (due to the momentum target generation); however, we show that it is not. Although SiMT increases the total training time of MAML by roughly 1.2 times, we have observed that it is 3 times faster to achieve the best performance of MAML: in Figure 3a, we compare the accuracy under the same training wall-clock time with MAML.

**Comparison of the momentum network and meta-model.** To understand how the momentum network improves the performance of the meta-model, we compare the adaptation performance of the momentum network and the meta-model during training SiMT. As shown in Figure 3b, we observe

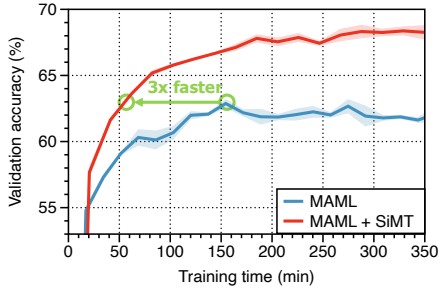

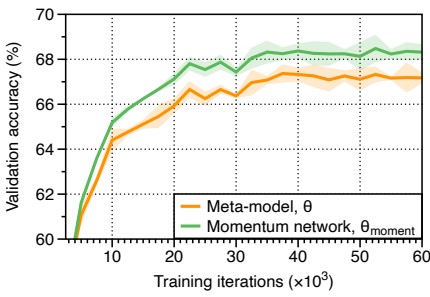

(a) Computation efficiency

(b) Network choice for the adaptation

Figure 3: Validation accuracy curves of 5-shot mini-ImageNet on Conv4: we compare the adaptation performance of (a) MAML and SiMT, under the same training wall-clock time, and (b) the meta-model and the momentum network of SiMT, under the same number of training steps. The solid line and shaded regions represent the mean and standard deviation, respectively, across three runs.

that the performance of the momentum network is consistently better than the meta-model, which implies that the proposed momentum target is a nice target model in our self-improving mechanism.

**Loss landscape analysis.** We visualize the loss landscape of the momentum network $\theta_{\texttt{moment}}$ and the meta-model $\theta$, to give insights into the generalization improvement. To do this, we train MAML with a momentum network (without distillation and dropout) and visualize the loss by perturbing each parameter space [29] (See Appendix D.2 for the detail of the visualization method). As shown in Figure 4, the momentum network forms a flatter loss landscape than the meta-model, where recent studies demonstrate that such a flat landscape is effective under various domains [12].

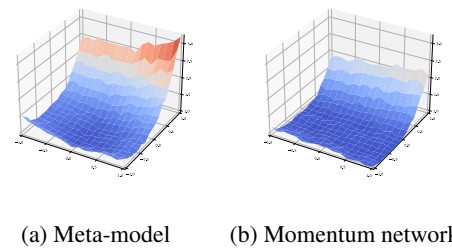

(a) Meta-model    (b) Momentum network

Figure 4: Loss landscape visualization of Conv4 trained on 5-shot mini-ImageNet with MAML.

## 6   Discussion and conclusion

In this paper, we propose a simple yet effective method, SiMT, for improving meta-learning. Our key idea is to efficiently generate target models using a momentum network and utilize its knowledge to self-improve the meta-learner. Our experiments demonstrate that SiMT significantly improves the performance of meta-learning methods on various applications.

**Limitations and future work.** While SiMT is a compute-efficient way to use target models in meta-learning, it is still built on top of existing meta-model update techniques. Since existing meta-learning methods have limited scalability (to large-scale scenarios) [44], SiMT is no exception. Hence, improving the scalability of meta-learning schemes is an intriguing future research direction, where we believe incorporating SiMT into such scenarios is worthwhile.

**Potential negative impacts.** Meta-learning often requires a large computation due to the numerous task adaptation during meta-training, therefore raising environmental concerns, e.g., carbon generation [41]. As SiMT is built upon the meta-learning methods, practitioners may need to consider some computation for successful training. To address this issue, sparse adaptation scheme [42] or lightweight methods for meta-learning [28] would be required for the applications.

## Acknowledgements

We thank Younggyo Seo, Jaehyun Nam, and Minseon Kim for providing helpful feedbacks and suggestions in preparing an earlier version of the manuscript. This work was supported by Institute of Information & communications Technology Planning & Evaluation (IITP) grant funded by the Korea government (MSIT) (No.2019-0-00075, Artificial Intelligence Graduate School Program (KAIST), No.2019-0-01906, Artificial Intelligence Graduate School Program (POSTECH), and No.2022-0-00713, Meta-learning applicable to real-world problems).

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
