# Appendix

## Meta-Learning with Self-Improving Momentum Target

## A Overview of terminologies used in the paper

- **Meta-model** $\theta$. The meta-learner network, i.e., learns to generalize on new tasks.

- **Adaptation subroutine** $\texttt{Adapt}(\cdot, \cdot)$. Algorithm for adapting the meta-model into a task expert by using a given task dataset.

- **Support set** $\mathcal{S}$. A dataset sampled from a given task distribution that is used for the adaptation.

- **Query set** $\mathcal{Q}$. A dataset sampled from a given task distribution (that is disjoint with the support set) to evaluate the adaptation performance of the algorithm.

- **Task-specific solver** $\phi$. Network adapted from the meta-model using the support set by using the adaptation subroutine, i.e., $\texttt{Adapt}(\theta, \mathcal{S})$

- **Momentum network** $\theta_{\texttt{moment}}$. Temporal ensemble of the meta-model where we use the exponential moving average of the meta-model parameter to compute the momentum.

- **Momentum target** $\phi_{\texttt{moment}}$. Network adapted from the momentum network using the support set by using the adaptation subroutine, i.e., $\texttt{Adapt}(\theta_{\texttt{moment}}, \mathcal{S})$

## B Overview of adaptation subroutine algorithms

**MAML [10] and extensions.** MAML uses the adaptation subroutine of taking a fixed number of SGD on the support set $\mathcal{S}$, starting from the meta-model parameter $\theta$. Formally, for a given $\mathcal{S}$, MAML with one step SGD obtains the task-specific solver $\phi$ by minimizing the empirical loss $\mathcal{L}$, as

$$\phi \leftarrow \theta - \alpha \nabla_\theta \mathcal{L}(\theta, \mathcal{S}), \tag{7}$$

where $\alpha$ denotes the step size. Here, we assume the empirical loss $\mathcal{L}$ is averaged over the given set $\mathcal{S}$. One can easily extend Eq. (7) to obtain $\phi$ with more than one SGD step from the meta-model $\theta$. For the extension, MetaSGD [31] learns the step size $\alpha$ along with the meta-model parameter $\theta$, and ANIL [36] only adapts the last linear layer of the meta-model $\theta$ to obtain $\phi$.

**ProtoNet [45].** The aim of metric-based meta-learning is to perform a non-parametric classifier on top of the meta-model's embedding space $f_\theta$. Specifically, ProtoNet learns a metric space in which classification can be performed by computing distances to prototype vectors of each class, i.e., $c_i := \frac{1}{|\mathcal{S}_i|} \sum_{(x,y) \in \mathcal{S}_i} f_\theta(x)$ where $\mathcal{S}_i$ contains the samples with class $i$ in the support set $\mathcal{S}$. Formally, for a given distance function $d(\cdot, \cdot)$, e.g., $l_2$ distance, the task-specific solver $\phi$ of the ProtoNet is as

$$p_\phi(y = i | x) = \frac{\exp(-d(f_\theta(x), c_i))}{\sum_{i'} \exp(-d(f_\theta(x), c_{i'}))}. \tag{8}$$

## C Application to reinforcement learning

We consider a reinforcement learning (RL) framework where an agent interacts with an environment in discrete time [48]. At each timestep $t$, the agent receives a state $\mathbf{s}_t$ from the environment and chooses an action $\mathbf{a}_t$ based on its policy $\pi(\mathbf{a}_t | \mathbf{s}_t)$. Then the environment gives a reward $r(\mathbf{s}_t, \mathbf{a}_t)$ and the agent transitions to the next state $\mathbf{s}_{t+1}$. The return $\mathcal{R} = \sum_{t=0}^{\infty} \gamma^t r(\mathbf{s}_t, \mathbf{a}_t)$ is defined as discounted cumulative sum of the reward with discount factor $\gamma \in [0, 1)$. As the goal of RL is to train a policy that maximizes the expected return, the loss is defined as a negative expected return as

$$\mathcal{L}(\theta) = - \mathop{\mathbb{E}}_{\mathbf{s}_t, \mathbf{a}_t \sim \pi_\theta} \left[ \sum_{t=0}^{\infty} \gamma^t r(\mathbf{s}_t, \mathbf{a}_t) \right], \tag{9}$$

where $\theta$ denotes the parameters of the policy. During meta-learning, the empirical version of this loss can be obtained using either the support set $\mathcal{S}^{\tau_i} = \{\mathbf{s}_t, \mathbf{a}_t \sim \pi_\theta\}$ in the adaptation subroutine, or the query set $\mathcal{Q}^{\tau_i} = \{\mathbf{s}_t, \mathbf{a}_t \sim \pi_{\phi^{\tau_i}}\}$ in the meta-update.

---

**Algorithm 2** Self-Improving Momentum Target for Reinforcement Learning

---

**Require:** Distribution over tasks $p(\tau)$, adaptation subroutine `Adapt`$(\cdot)$, momentum coefficient $\eta$,
      weight hyperparameter $\lambda$, task batch size $N$, number of rollouts per task $K$, learning rate $\beta$.

---

1: Initialize $\theta$ using the standard initialization scheme.
2: Initialize the momentum network with the meta-model parameter, $\theta_{\texttt{moment}} \leftarrow \theta$.
3: **while** not done **do**
4:     Sample $N$ tasks $\{\tau_i\}_{i=1}^{N}$ from $p(\tau)$
5:     **for** $i = 1$ to $N$ **do**
6:         Sample $K$ trajectories $\mathcal{S}^{\tau_i}$ using $\pi_\theta$ from $\tau_i$
7:         $\phi_{\texttt{moment}}^{\tau_i} = \texttt{Adapt}(\theta_{\texttt{moment}}, \mathcal{S}^{\tau_i})$.                 ▷ Generate a momentum target.
8:         $\phi^{\tau_i} = \texttt{Adapt}(\theta, \mathcal{S}^{\tau_i})$.                        ▷ Adapt a task-specific solver.
9:         Sample trajectories $\mathcal{Q}^{\tau_i}$ using $\pi_{\phi^{\tau_i}}$ from $\tau_i$
10:       $\mathcal{L}_{\texttt{total}}^{\tau_i}(\theta) = (1-\lambda) \cdot \mathcal{L}_{\texttt{TRPO}}^{\tau_i} + \lambda \cdot \mathcal{L}_{\texttt{teach}}(\phi^{\tau_i}, \phi_{\texttt{moment}}^{\tau_i}, \mathcal{Q}^{\tau_i})$         ▷ Compute loss.
11:     **end for**
12:     $\theta \leftarrow \theta - \frac{\beta}{N} \cdot \nabla_\theta \sum_{i=1}^{N} \mathcal{L}_{\texttt{total}}^{\tau_i}(\theta)$.             ▷ Train the meta-model.
13:     $\theta_{\texttt{moment}} \leftarrow \eta \cdot \theta_{\texttt{moment}} + (1-\eta) \cdot \theta$.         ▷ Update the momentum network.
14: **end while**

---

In meta-RL setup, each task $\tau_i$ contains an initial state distribution $q_i(\mathbf{s}_0)$ and a transition distribution $q_i(\mathbf{s}_{t+1}|\mathbf{s}_t, \mathbf{a}_t)$. The goal of meta-RL is to optimize a policy $\pi_\theta$ that minimizes $\mathcal{L}(\theta)$ for unseen tasks $\tau_i \sim p(\tau)$ using only a limited number ($K$) of sampled trajectories. In Algorithm 2, we describe the meta-RL algorithm with our proposed method, SiMT. We note that SiMT can be combined with any (gradient-based) meta-learning approaches and policy gradient methods. In our experiments, we built SiMT upon MAML with vanilla policy gradient [59] and trust-region policy optimization (TRPO; [39]) for the task-specific solver and meta-model, respectively, as we described in Section 5.3.

Here, we describe the detailed objective of the meta-RL used in the experiments. To this end, we use the following standard definitions of the state-action value function $Q^\pi$, the value function $V^\pi$, and the advantage function $A^\pi$.

$$Q^\pi(\mathbf{s}_t, \mathbf{a}_t) = \underset{\mathbf{s}_{t+1}, \mathbf{a}_{t+1}, \ldots}{\mathbb{E}}\left[\sum_{k=0}^{\infty} \gamma^k r(\mathbf{s}_{t+k})\right], \; V^\pi(\mathbf{s}_t) = \underset{\mathbf{a}_t, \mathbf{s}_{t+1}, \ldots}{\mathbb{E}}\left[\sum_{k=0}^{\infty} \gamma^k r(\mathbf{s}_{t+k})\right],$$
$$A^\pi(\mathbf{s}_t, \mathbf{a}_t) = Q^\pi(\mathbf{s}_t, \mathbf{a}_t) - V^\pi(\mathbf{s}_t), \; \text{where} \; \mathbf{a}_t \sim \pi(\mathbf{a}_t|\mathbf{s}_t), \mathbf{s}_{t+1} \sim q(\mathbf{s}_{t+1}|\mathbf{s}_t, \mathbf{a}_t). \quad (10)$$

For better clarity, we will omit the notation $t$ if there is no confusion. The gradient of $\mathcal{L}(\theta, \mathcal{S})$ obtained by the vanilla policy gradient method is

$$\nabla_\theta \mathcal{L}(\theta, \mathcal{S}) = -\frac{1}{|\mathcal{S}|} \sum_{\mathbf{s}, \mathbf{a} \in \mathcal{S}} \nabla_\theta \log \pi_\theta(\mathbf{a}|\mathbf{s}) A^{\pi_\theta}(\mathbf{s}, \mathbf{a}), \quad (11)$$

where $\mathcal{S}$ is a set of trajectories sampled using the policy $\pi_\theta$. Then the adaptation subroutine of the parameters $\theta$ is performed as follows: $\phi = \texttt{Adapt}(\theta, \mathcal{S}) := \theta - \alpha \cdot \nabla_\theta \mathcal{L}(\theta, \mathcal{S})$. For SiMT, we update the momentum network $\theta_{\texttt{moment}}$ by using the trajectories $\mathcal{S}$ from the policy $\pi_\phi$ (line 7 in Algorithm 2), since it performs better and efficient than using additional trajectories from $\pi_{\phi_{\texttt{moment}}}$.

In the meta-update procedure, we use the surrogate objective of TRPO to update the parameters $\theta$. Let $\theta_{\text{old}}$ and $\theta$ denote parameters of current and new policies (for every meta-update iteration of Algorithm 2), respectively. Then the theoretical TRPO update is

$$\theta \leftarrow \underset{\theta}{\arg\min} \; \frac{1}{N} \sum_{i=1}^{N} \mathcal{L}_{\texttt{TRPO}}^{\tau_i}(\theta_{\text{old}}, \theta) \; \text{subject to} \; \overline{\text{KL}}(\theta_{\text{old}}||\theta) \leq \delta, \quad (12)$$

where $\mathcal{L}_{\texttt{TRPO}}(\theta_{\text{old}}, \theta)$ is the (negative) surrogate advantage from adapted parameters $\phi$ and $\phi_{\text{old}}$, a measure of how the policy $\pi_\phi$ performs relative to the old policy $\pi_{\phi_{\text{old}}}$, using trajectories $\mathcal{Q}$ which is

sampled from the old policy:

$$\mathcal{L}_{\text{TRPO}}(\theta_{\text{old}}, \theta) = -\frac{1}{|\mathcal{Q}|} \sum_{\mathbf{s}, \mathbf{a} \in \mathcal{Q}} \frac{\pi_\phi(\mathbf{a}|\mathbf{s})}{\pi_{\phi_{\text{old}}}(\mathbf{a}|\mathbf{s})} A^{\pi_{\phi_{\text{old}}}}(\mathbf{s}, \mathbf{a}), \tag{13}$$

and $\overline{\text{KL}}(\theta||\theta_{\text{old}})$ is an average KL divergence between policies across states visited by the old policy

$$\overline{\text{KL}}(\theta_{\text{old}}||\theta) = \frac{1}{N} \sum_{i=1}^{N} \frac{1}{|\mathcal{Q}^{\tau_i}|} \sum_{\mathbf{s}, \mathbf{a} \in \mathcal{Q}^{\tau_i}} \left[ \text{KL}\left( \pi_{\phi_{\text{old}}^{\tau_i}}(\cdot|\mathbf{s}) || \pi_{\phi^{\tau_i}}(\cdot|\mathbf{s}) \right) \right]. \tag{14}$$

To utilize SiMT in meta-RL tasks, we define the knowledge distillation loss by using the task-specific policies $\pi_\phi$ and $\pi_{\phi_{\text{moment}}}$, which are parameterized by $\phi$ and the momentum target $\phi_{\text{moment}}$, respectively. For a given trajectories $\mathcal{Q}$ sampled using $\pi_\phi$, the knowledge distillation loss is as

$$\mathcal{L}_{\text{teach}}(\phi, \phi_{\text{moment}}, \mathcal{Q}) := \frac{1}{|\mathcal{Q}|} \sum_{\mathbf{s}, \mathbf{a} \in \mathcal{Q}} \text{KL}\big( \pi_\phi(\mathbf{s}) \, || \, \pi_{\phi_{\text{moment}}}(\mathbf{s}) \big). \tag{15}$$

For meta-learning with SiMT, the objective $\mathcal{L}_{\text{TRPO}}^{\tau_i}$ in (12) is replaced with $\mathcal{L}_{\text{total}}^{\tau_i}$ in Algorithm 2.

# D   Experimental details

In this section, we provide the experimental details, including experimental setups and loss landscape visualization. The implementation of all experiments is given in the supplementary material.

## D.1   Experimental setup details

**Network architecture details.** For all few-shot classification experiments, we mainly follow the setups from MAML [10] and also consider the setups from recent papers [33, 56]. The Conv4 classifier consists of four convolutional layers, each with 64 filters, followed by a batch normalization (BN) layer [22] as well as a max-pooling layer with kernel size and stride of 2. The network then projects to its output through a linear layer. We choose to use the 64 channels for each layer by following the recent papers, e.g., ANIL, BOIL [33], and sparse-MAML [56]. For the ResNet-12, we use the residual blocks with channel sizes of 64, 128, 256, and 512, where the architecture is identical to the one used in previous meta-learning studies [34, 33]. For SiMT, we apply dropout before the max-pooling layer. We also observed that the Dropblock regularization [15] (commonly used in ResNet architectures in few-shot classification) shows a similar performance gain as the dropout in ResNet-12. We use additional three convolutional layers with 64 channels in front of Conv4 architecture for all few-shot regression tasks by following the prior works [63, 13].

In the meta-RL setup, the policy is a conditional probability distribution $\pi_\theta(\mathbf{a}|\mathbf{s})$ parameterized by a neural network $\theta$. For a given state vector $\mathbf{s}$, this neural network specifies a distribution over action space. Then one can compute the likelihood $q(\mathbf{a}|\mathbf{s})$ and sample the action $\mathbf{a} \sim q(\cdot|\mathbf{s})$. For our experiments with continuous action spaces, we use a Gaussian distribution, where the covariance matrix is diagonal and independent of the state. Specifically, the policy has a multi-layer perceptron (MLP) which computes the mean and a learnable vector for log standard deviation with the same dimension as $\mathbf{a}$. We use a 2-layer MLP with hidden dimensions of 100 and ReLU activation for the policy, as we described in Section 5.3.

**Training details.** For gradient-based few-shot classification experiments, we mainly follow MAML [10] and partially follow Von Oswald et al. [56] (e.g., ResNet-12 training setups). For ProtoNet, we follow a recent paper by Yao et al. [61] on Conv4 and extend the setup into ResNet-12 by only changing the training iterations. We follow a recent paper by Gao et al. [13] for few-shot regression tasks. We use Adam optimizer [25] for optimizing the meta-model $\theta$ and train 60,000 iterations for Conv4 (including regression models), and 30,000 iterations for ResNet-12. When training, we use the learning rate of $\beta = 1\text{e}{-3}$ for classification and $\beta = 5\text{e}{-4}$ for regression. As for the details of MAML and ANIL, we use 5 step SGD with a fixed step size $\alpha = 1\text{e}{-2}$ for the classification and $\alpha = 2\text{e}{-3}$ for the regression. For all few-shot learning, we use 1 step SGD on MetaSGD. For the classification, we use a task batch size of 4 and 2 tasks for 1-shot and 5-shot gradient-based methods, respectively, and use 1 for the rest. For the regression, we use a task batch size of 10. Note that all adaptation hyperparameters for the task-specific solver $\phi$ and momentum target $\phi_{\text{moment}}$ are the same.

Moreover, we handle BN parameters following the transductive learning setting for gradient-based approaches and use inductive BN parameters for ProtoNet.

For meta-RL experiments, we mainly follow the setups from MAML [10] and use the open-source PyTorch implementation [1] of MAML for RL. To optimize the objective of TRPO (and SiMT), we compute the Hessian-vector products to avoid computing third derivatives and use a line search to ensure improvement of the surrogate objective with the satisfaction of the KL divergence constraint. For both learning and meta-learning updates, we use the standard linear feature baseline proposed by Duan et al. [8]. To estimate the advantage function, we use a generalized advantage estimator (GAE; [40]) with a discount factor of 0.95 and a bias-variance tradeoff of 1.0. In all meta-RL experiments, the policy is trained using a single gradient step with $\alpha = 0.1$, with rollouts $K = 20$ per gradient step. We use a meta batch size $N$ of 20 and 40 for the 2D navigation and the locomotion tasks, respectively. For TRPO, we use $\delta = 0.01$ for all experiments. The agent is meta-updated for 500 iterations, and the model with the best average return during training is used for the evaluation.

**Hyperparameter details for SiMT.** SiMT requires hyperparameters, including weight hyperparameter $\lambda$, momentum coefficient $\eta$, and the dropout probability $p$.

We first provide the hyperparameter details of few-shot learning. For the momentum coefficient $\eta$, we use $0.995$ except for the 5-shot classification, where $0.999$ shows slightly better performance. For the weight hyper-parameter $\lambda$, we use $0.5$ for MAML and ANIL, $0.1$ for MetaSGD, and $5e - 3$ for ProtoNet, respectively. Finally, for the dropout probability $p$, we use $0.2$ for MAML and ANIL, and $0.1$ for the rest. Also, we find that the temperature value of $T = 4$ works the best as in the prior knowledge distillation works [65, 32].

For meta-RL experiments, we use the momentum coefficient of $0.995$ for 2D navigation and $0.99$ for locomotion tasks, respectively. For the weight hyperparameter $\lambda$, we use $0.1$ for 2D navigation and $0.02$ for locomotion tasks, respectively. We linearly ramp up $\lambda$ for locomotion tasks, as the momentum target might poorly perform at the beginning of the training.

**Dataset and environment details.** For the few-shot regression task, we use Pascal [63], which contains 65 objects from 10 categories. We sample 50 objects for training and the other 15 objects for testing. $128 \times 128$ gray-scale images are rendered for each object with a random rotation in azimuth angle normalized between $[0, 10]$. We also use ShapeNet [13], which includes 30 categories. 27 of these are used during training and the other three categories are used for the evaluation.

For few-shot classification, we use four image datasets, including mini-ImageNet [55, 37], tiered-ImageNet [38], CUB [57], and Cars [26]. We consider a 5-way classification for all tasks by following the prior works [10, 33]. The mini-ImageNet consists of training, validation, and test sets with 64, 16, and 20 classes in each, respectively. The tiered-ImageNet consists of datasets with 351 training, 97 validation, and 160 test classes. By following Hilliard et al. [20], we split CUB dataset into 100 training, 50 validation, and 50 test classes. Cars are split into training, validation, and test sets with 98, 49, and 49 classes in each, respectively, by following Tseng et al. [54].

For meta-RL experiments, we consider 2D navigation and Half-cheetah locomotion tasks, which were considered in previous works [10, 36]. In 2D navigation tasks, the observation is the current position in a 2D unit square, and the action is a velocity command which clipped in the range of $[-0.1, 0.1]$. The reward is the negative squared distance to the goal, and episodes terminate when the agent is within 0.01 of the goal or arrives at the horizon. In Half-cheetah goal direction tasks, the reward is the magnitude of the velocity in either the forward or backward direction, randomly chosen for each task. The horizon $H$ is 100 and 200, for 2D navigation and locomotion tasks, respectively.

**Evaluation details.** All our few-shot learning results are reported for models that are early-stopped by measuring the average validation set accuracy (across 2,000 validation set tasks for classification and 100 validation set tasks for regression). For the test accuracy, we report the mean accuracy of 2,000 test tasks and 100 test tasks for classification and regression, respectively. By following the original papers, we use 10 SGD steps for the adaptation on MAML and ANIL, and use 1 SGD step on MetaSGD. For meta-RL experiments, the step size during adaptation was set to $\alpha = 0.1$ for the first step, and $\alpha = 0.05$ for all future steps, following the evaluation setup of MAML. We report the truncated mean and standard deviation of the performance using five random seeds, i.e., the statistics after discarding the best and worst seeds, for meta-RL experiments.

Table 7: Few-shot image classification accuracy (%) on ResNet-10 trained with mini-ImageNet. We consider in-domain and cross-domain scenarios where CUB, Cars, Places, and Plantae are used as cross-domain datasets. SiMT utilizes the momentum network for the adaptation. The reported results are 95% confidence intervals averaged over 1,000 meta-test tasks, subscripts denote the standard deviation, and bold denotes the best result of each group. * indicates the values from the reference.

| Problem | Method | mini-ImageNet $\rightarrow$ | | | | |
| | | mini-ImageNet | CUB | Cars | Places | Plantae |
| --- | --- | --- | --- | --- | --- | --- |
| 1-shot | GNN [14]* | $60.77_{\pm 0.75}$ | $45.69_{\pm 0.68}$ | $31.79_{\pm 0.51}$ | $53.10_{\pm 0.80}$ | $35.60_{\pm 0.56}$ |
| | GNN [14] + FT [54]* | $66.32_{\pm 0.80}$ | $47.47_{\pm 0.75}$ | $31.61_{\pm 0.53}$ | $55.77_{\pm 0.79}$ | $35.95_{\pm 0.58}$ |
| | GNN [14] + SiMT | $67.22_{\pm 0.79}$ | $48.19_{\pm 0.71}$ | $32.47_{\pm 0.57}$ | $57.41_{\pm 0.81}$ | $\mathbf{38.13_{\pm 0.61}}$ |
| | GNN [14] + FT [54] + SiMT | $\mathbf{68.02_{\pm 0.80}}$ | $\mathbf{48.75_{\pm 0.76}}$ | $\mathbf{32.89_{\pm 0.69}}$ | $\mathbf{58.23_{\pm 0.86}}$ | $38.07_{\pm 0.60}$ |
| 5-shot | GNN [14]* | $80.87_{\pm 0.56}$ | $62.25_{\pm 0.65}$ | $44.28_{\pm 0.63}$ | $70.84_{\pm 0.65}$ | $52.53_{\pm 0.59}$ |
| | GNN [14] + FT [54]* | $81.98_{\pm 0.55}$ | $66.98_{\pm 0.68}$ | $44.90_{\pm 0.64}$ | $73.94_{\pm 0.67}$ | $53.85_{\pm 0.62}$ |
| | GNN [14] + SiMT | $84.37_{\pm 0.56}$ | $68.78_{\pm 0.69}$ | $45.61_{\pm 0.67}$ | $76.73_{\pm 0.66}$ | $55.72_{\pm 0.63}$ |
| | GNN [14] + FT [54] + SiMT | $\mathbf{85.13_{\pm 0.55}}$ | $\mathbf{70.09_{\pm 0.67}}$ | $\mathbf{46.90_{\pm 0.65}}$ | $\mathbf{78.15_{\pm 0.64}}$ | $\mathbf{56.60_{\pm 0.64}}$ |

**Resource details.** For estimating the training time of our method (in Section 2, Section 5.5, and Appendix E.3), we use the same machine and stop other processes: Intel(R) Xeon(R) Silver 4214 CPU @ 2.20GHz and a single RTX 2080 Ti GPU for the measurement. For the main development, we mainly use Intel(R) Xeon(R) Gold 6226R CPU @ 2.90GHz and a single RTX 3090 GPU.

## D.2 Loss landscape visualization

In this subsection, we provide the loss landscape visualization details of Figure 4. For the experiment, we consider vanilla MAML $\theta$ and its momentum network $\theta_{\texttt{moment}}$ without distillation loss and dropout regularization. We change the parameter by adding two direction vectors $d_1, d_2$ and visualize the resulting loss value. Specifically, for a given task distribution of training set $p(\tau)$, a training loss $\mathcal{L}_{\texttt{train}}(\theta) := \mathbb{E}_{\tau \sim p(\tau)} \big[ \mathcal{L}(\texttt{Adapt}(\theta, \mathcal{S}^\tau), \mathcal{Q}^\tau) \big]$, and the center point $\theta$ (or $\theta_{\texttt{moment}}$), we visualize

$$\mathcal{L}_{\texttt{train}}(\theta + i \cdot d_1 + j \cdot d_2), \tag{16}$$

where $i, j$ are the step size of the visualization axis. For choosing visualization directions, we sample two random vectors from a unit Gaussian distribution and normalize each filter in the vector to have the same norm of the corresponding filter in $\theta$ by following the prior work [29].

# E More experimental results

## E.1 Comparison with another meta-learning regularization method

We also compare SiMT with a recent regularization method, feature-wise transformation (FT) [54], a regularization method for metric-based meta-learning schemes. FT augments the image features by utilizing the affine transformation in the feature layer to capture the data distribution. While FT shows strong benefits in improving the base meta-learner, it focuses on cross-domain generalization and requires multiple datasets to learn the hyperparameters. In this regard, we consider a more generic scenario and compare SiMT with FT on a standard few-shot classification setup in [54][3]. Here, we train ResNet-10 on the mini-ImageNet dataset and use GNN [14] as a base meta-learning scheme, which shows the best performance when used with FT. Then, we evaluate both methods on few-shot in-domain and cross-domain adaptation scenarios. As shown in Table 7, SiMT consistently outperforms FT in all cases. More importantly, we observed that SiMT and FT have an orthogonal benefit where joint usage further improves the performance.

## E.2 Momentum targets for a small proportion of tasks

To further investigate the efficacy of the target model, we consider generating target models for a small proportion of tasks when training SiMT. To this end, we train Conv4 on a mini-ImageNet dataset and control the ratio of tasks with the target model. As shown in Table 8, SiMT shows

---

[3]We use the official implementation from `https://github.com/hytseng0509/CrossDomainFewShot`

Table 8: Few-show in-domain adaptation accuracy (%) of SiMT under different proportion of tasks with momentum targets. We train Conv4 under 5-way mini-ImageNet dataset. Note that all model utilizes the momentum network for the adaptation. The reported results are averaged over three trials, subscripts denote the standard deviation, and bold indicates the best result of each group. $p$ indicates the proportion of tasks with momentum targets.

| $p$ | 0% | 5% | 10% | 25% | 50% | 100% |
|---|---|---|---|---|---|---|
| 1-shot | $48.98_{\pm0.32}$ | $49.83_{\pm0.12}$ | $49.92_{\pm0.23}$ | $50.42_{\pm0.67}$ | $50.55_{\pm0.41}$ | $\mathbf{51.49_{\pm0.18}}$ |
| 5-shot | $66.12_{\pm0.21}$ | $66.81_{\pm0.26}$ | $66.99_{\pm0.24}$ | $67.15_{\pm0.18}$ | $67.49_{\pm0.39}$ | $\mathbf{68.74_{\pm0.12}}$ |

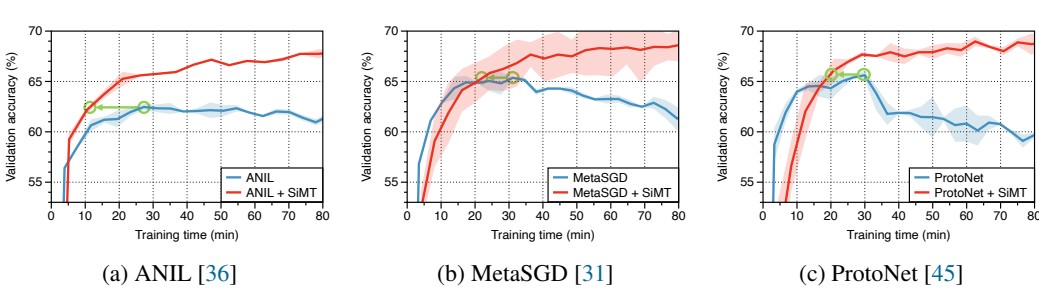

(a) ANIL [36]      (b) MetaSGD [31]      (c) ProtoNet [45]

Figure 5: Comparison of the computational efficiency between the base adaptation subroutine algorithm and SiMT: we compare the accuracy of mini-ImageNet 5-shot classification (on Conv4) under the same training wall-clock time. The solid line and shaded regions represent the mean and standard deviation, respectively, across three runs.

Table 9: Few-shot in-domain adaptation accuracy (%) on additional adaptation subroutine algorithms. We train Conv4 under 5-way mini- and tiered-ImageNet. SiMT utilizes the momentum network for the adaptation. The reported results are averaged over three trials, subscripts denote the standard deviation, and bold denotes the best result of each group.

| | mini-ImageNet | | tiered-ImageNet | |
|---|---|---|---|---|
| Method | 1-shot | 5-shot | 1-shot | 5-shot |
| FOMAML [10] | $45.96_{\pm0.61}$ | $62.58_{\pm0.54}$ | $47.85_{\pm0.46}$ | $64.21_{\pm0.50}$ |
| FOMAML [10] + SiMT | $\mathbf{47.78_{\pm0.57}}$ | $\mathbf{65.79_{\pm0.03}}$ | $\mathbf{48.55_{\pm0.08}}$ | $\mathbf{65.79_{\pm0.52}}$ |
| BOIL [33] | $49.78_{\pm0.65}$ | $66.98_{\pm0.41}$ | $52.19_{\pm0.46}$ | $68.88_{\pm0.43}$ |
| BOIL [33] + SiMT | $\mathbf{50.83_{\pm0.09}}$ | $\mathbf{67.77_{\pm0.24}}$ | $\mathbf{52.44_{\pm0.28}}$ | $\mathbf{69.05_{\pm0.27}}$ |

consistent improvement even with a small portion of task models, e.g., 5% of target models show 1% improvement in 1-shot adaptation. However, remark that one can easily generate target models for all tasks with SiMT, hence, one does not need to use few targets in practice.

### E.3   Computation efficiency of SiMT

To further analyze the computational efficiency of SiMT, we consider additional comparisons on meta-learning methods, including ANIL [36], MetaSGD [31], and ProtoNet [45] (we train Conv4 on the mini-ImageNet dataset under a 5-shot classification scenario). Here, we also observed consistent efficiency gain on these setups, e.g., 30% of training time reduced to achieve the peak accuracy of ProtoNet when using SiMT. Although SiMT is slower than the base algorithms when comparing the number of optimization steps (due to the momentum target generation), we want to note that SiMT can be more efficient in terms of time to reach the same performance.

### E.4   Additional adaptation subroutine algorithms

We consider more adaptation subroutine algorithms to verify the effectiveness of SiMT: first-order approximation of MAML (FOMAML) [10] and BOIL [33]. FOMAML removes the second derivatives of MAML, and BOIL does not adapt the last linear layer of the meta-model $\theta$ under MAML. As shown in Table 9, SiMT consistently improves both meta-learning schemes in all tested cases.