# OpenReview forum: "Meta-Learning with Self-Improving Momentum Target"
_NeurIPS.cc/2022/Conference — NeurIPS 2022 Accept_

### Official Review · Reviewer_mjLb · 2022-07-09

**Rating:** 7
**Confidence:** 4
**Soundness:** 4 excellent
**Presentation:** 4 excellent
**Contribution:** 3 good

**Summary:**

The paper introduces a new method for few-shot learning that utilises a self-improving momentum target. The idea is to learn from a meta-model that is more quickly adapted than the standard meta-model - the momentum target is updated using exponential moving average rather than SGD. The approach can be combined with a wide variety of few-shot learners and consistently leads to strong improvements in performance.

**Questions:**

* Based on Figure 3b, it seems that even just using momentum network instead of the meta-model leads to strong performance. Is it the case that if we simply replace the outer loop SGD update with exponential moving average in MAML it would significantly improve its performance? Is it something which any other few-shot learners already do?
* How easy is it to select the additional hyperparameters for SiMT?


**Limitations:**

Yes, the discussion of limitations and societal impacts is well-written.

**Strengths And Weaknesses:**

Strengths:
* The approach can be combined with various few-shot learning methods and consistently leads to strong improvements in performance.
* The idea itself is intuitive and novel, even if the idea has some similarities to bootstrapped meta-learning and momentum networks
* The paper provides excellent explanations and illustrations to show how the method works
* The empirical evaluation is extensive and covers few-shot regression, few-shot classification (both in domain and across domains) and also reinforcement learning - with strong results across all
* There is an ablation study evaluating the various components showing that all parts are important

Weaknesses:
* There are quite many additional components in the approach
* The approach introduces several additional hyperparameters for which we need to select values (but perhaps selecting them is not too difficult)

I’ve enjoyed reading the paper and the strengths significantly outweigh the weaknesses, so I recommend acceptance (the weaknesses are rather minor). The paper will likely be interesting and valuable for the ML community, especially for the researchers focused on meta-learning.

---

> ### Author Response · Authors · 2022-08-02
> **Response to Reviewer mjLb**
>
> Dear reviewer mjLb,
>
> We sincerely appreciate your efforts and insightful comments to improve the manuscript. We respond to each of your comments one-by-one in what follows. In the revised draft, we mark our revisions by “blue”.
>
> ------------------------------------------------
>
> **[W1] Many additional components in the method.**
>
> Our method might seemingly have many components, but in the perspective of efficient target model generation, the components are not so many: our major component is to use the momentum network, and dropout is used for stabilizing the self-improving process when using the momentum target.
>
> ------------------------------------------------
>
> **[W2 & Q2] Introduce some additional hyperparameters**
>
> While our method consists of additional hyperparameters, we found that the values are somewhat easy to select. In particular, we found that the commonly used hyperparameter values in the other literature also perform well on SiMT, e.g., high momentum coefficient $\eta$ = 0.995, 0.999 [1,2]. Moreover, we found that hyperparameter selection is not sensitive across datasets or architectures and only varies depending on the base meta-learning scheme.
>
> [1] He et al., Momentum Contrast for Unsupervised Visual Representation Learning, CVPR 2020\
> [2] Grill et al., Bootstrap your own latent: A new approach to self-supervised Learning, NeurIPS 2020
>
> ------------------------------------------------
>
> **[Q1] Result of replacing the outer loop SGD update with exponential moving average (EMA).**
>
> We kindly remind that the result of using EMA on the outer loop of MAML can be found in Table 4 (we also give part of the table below). To our knowledge, EMA on the outer loop of meta-learning is not commonly used in the meta-learning literature, where SiMT utilizes it to efficiently generate the target model. During the project, we also tried using EMA on the inner loop of MAML, where it showed no improvement. We conjecture this is because EMA requires many gradient steps (to update the parameter) while MAML only uses a few gradient steps for the inner loop. We believe using EMA on the multiple inner-loop gradient steps [1] can be an interesting future direction.
>
> \begin{array}{lcccc}
> \hline
> \text{Method}  & \text{1-shot} & \text{5-shot} \newline
> \hline
> \text{MAML} & 47.33\small{\pm{0.45}} & 63.27\small{\pm{0.14}} \newline
> \text{MAML + momentum}  & 48.98\small{\pm{0.32}} & 66.12\small{\pm{0.21}} \newline
> \hline
> \end{array}
>
> [1] Nichol et al., On First-Order Meta-Learning Algorithms, arXiv 2016

---

> > ### Comment · Reviewer_mjLb · 2022-08-04
> > **Thank you**
> >
> > Thank you for these additional explanations, it is valuable to know answers to the questions I had. Also after reading the other reviews I still think it is a good paper that will be of interest to the ML community, so I keep my rating.

---

> > > ### Author Response · Authors · 2022-08-04
> > > **Thank you for the response.**
> > >
> > > Thank you for letting us know! We are happy to hear that our rebuttal addressed your questions well.
> > >
> > > If you have any further questions or suggestions, please do not hesitate to let us know.
> > >
> > > Thank you very much,\
> > > Authors

---

### Official Review · Reviewer_siDw · 2022-07-10

**Rating:** 6
**Confidence:** 4
**Soundness:** 3 good
**Presentation:** 2 fair
**Contribution:** 2 fair

**Summary:**

An algorithm based on knowledge distillation together with a momentum network working as experts on the unseen fields to solve meta-learning problems. The proposed algorithm shows compatibility with multiple existing meta-models and improvement. The general idea is simple and easy to implement.

**Questions:**

1. It seems like the main idea of the proposed method is to stitch a momentum network and knowledge distillation. What is the motivation for doing this? I think one thing could be good to show is that how the momentum network works on each task after the adaptation, since the knowledge is distilled from it.
2. The adapted moving average parameter set, momentum, is adapted to the query set, to be an expert. Is it similar to say that the whole idea is to regularise the meta-model to its momentum model?
3. Why the perturb (dropout) is important? Without dropout how does the model perform differently?

**Limitations:**

The paper is presented in an unfriendly way for the readers who do not have an efficient background. In the beginning, especially in the introduction section, a lot of concepts, target model, original model and task-specific solvers, are introduced in the cause levels making the paper very confusing. Some components of the whole algorithm are explained very independently. It is not very clear why one relies on the other.

**Strengths And Weaknesses:**

Strengths:
1. The experiment shows the good performance of the model on a variety of tasks.
2. The model is analysed from different perspectives, loss landscape, and computational efficiency.

Weakness:
1. From my understanding, despite all the complicated presentation about all the algorithm components, the algorithm only forces the meta-model to get close to its momentum model which works as a regulariser. From this perspective, the novelty is quite limited.
2. The proposed algorithm shows the improvement and compatibility on multiple MAML models but lacks comparison with other SOTA meta-models.

---

> ### Author Response · Authors · 2022-08-02
> **Response to Reviewer siDw (3/3)**
>
> **[L1] More friendly writing for readers without efficient background.**
>
> Following the suggestion, we have clarified the confusing term in the introduction, e.g., original model $\to$ original meta-model. Furthermore, to help readers new to this field, we added a brief explanation of the terminology used in our paper in Appendix A. We hope this can guide the readers to understand our paper better.
>
> ------------------------------------------------
>
> **[Q1] Motivation for combining momentum network and knowledge distillation: “Good to show how the momentum network works after the adaptation”.**
>
> We first kindly remind that the task adaptation performance of the momentum network is better than the meta-model (line 47 and Table 4). Motivated by this, we suggest generating target models from the momentum network which can distill knowledge to the meta-model itself (line 59). Note that this improved meta-model recursively improves the momentum (through the temporal ensembling), hence, forming a self-improving process. Moreover, we even find that the momentum network consistently performs better than the meta-model during the training SiMT, demonstrating the self-improving mechanism truly works (Fig 3(b)).
>
> ------------------------------------------------
>
> **[Q2] Is it similar to say that the idea is to regularise the meta-model to its momentum model?**
>
> Yes, our methodology can also be interpreted as regularizing the meta-model with good knowledge of momentum networks. In detail, we generate a target model (i.e., a teacher) from the momentum network and distill the knowledge of the target model to the adapted student network from the meta-model.
>
> ------------------------------------------------
>
> **[Q3] Importance of perturbation (dropout)**
>
> We find that the distillation loss converges too fast during the training, which implies that the momentum network and the meta-model become too similar to each other (i.e., small distillation effect so it stops the self-improving process). To prevent such an issue, we suggest using dropout regularization on the student network (i.e., task-adapted model from the meta-model). Intuitively, injecting noise into the parameter space forces an asymmetricity between the student and the teacher’s prediction, hence, preventing two models from reaching a similar prediction. As shown in Table 4, dropout regularization effectively improves performance when it is jointly applied with the distillation, e.g., 66.52\% $\to$ 68.74\%.

---

> ### Author Response · Authors · 2022-08-02
> **Response to Reviewer siDw (2/3)**
>
> **[W2] Lacks comparison with other SOTA meta-models.**
>
> We would like to clarify that SiMT can be applied to any meta-learning method, hence, our focus was to empirically verify the compatibility of SiMT throughout the experiments rather than comparing it with SOTA methods. In this regard, we have rather compared SiMT with another regularization method in meta-learning in Appendix E.1: we provide an empirical comparison with a recent regularization method called Bootstrapped meta-learning (Bootstrap) [1]. SiMT consistently outperforms Bootstrap, as is shown in Table 5; we also give a part of the table below. We further note that SiMT has broader applicability than Bootstrap, as the latter is specialized for gradient-based meta-learning schemes, while SiMT can also be applied to other types of meta-learning, e.g., metric-based methods (such as ProtoNet).
>
> \begin{array}{lcccc}
> \hline
> \text{Method}  &\text{mini-ImageNet}  & & \text{tiered-ImageNet}  &  \newline
> & \text{1-shot} & \text{5-shot} & \text{1-shot} &  \text{5-shot} \newline
> \hline
> \text{MAML} & 47.33\small{\pm{0.45}} & 63.27\small{\pm{0.14}} & 50.19\small{\pm{0.21}} & 66.05\small{\pm{0.19}} \newline
> \text{MAML + Bootstrap [1]}  & 48.68\small{\pm{0.33}} & 68.45\small{\pm{0.40}} & 49.34\small{\pm{0.26}} & 68.84\small{\pm{0.37}} \newline
> \text{MAML + SiMT} & \mathbf{51.49\small{\pm{0.18}}} & \mathbf{68.74\small{\pm{0.12}}} & \mathbf{52.51\small{\pm{0.21}}} & \mathbf{69.58\small{\pm{0.11}}} \newline
> \hline
> \end{array}
>
> In addition, following the suggestion of Reviewer DCr7, we provide additional comparisons with the regularization method of [2], i.e., feature-wise transformation (FT) regularization (a transformation layer in the feature space to capture more diverse distributions). We consider the few-shot classification setup as in [1] and train ResNet-10 on the mini-ImageNet dataset, using GNN [3] as the base meta-learning method. As shown in the table, SiMT consistently outperforms FT in all cases. More importantly, we observe that SiMT and FT can be used jointly to improve the performance further. We have added this result to the revised manuscript (in Appendix E.1, Table 6).
>
> \begin{array}{llccccc}
> \hline
> \text{Problem}  & \text{Method}  &\text{mini-ImageNet}  & \text{CUB}  &  \text{Cars}& \text{Places} & \text{Plantae}  \newline
> \hline
> & \text{GNN} & 60.77\small{\pm{0.75}} & 45.69\small{\pm{0.68}} & 31.79\small{\pm{0.51}} & 53.10\small{\pm{0.80}} & 35.60\small{\pm{0.56}} \newline
> \text{1-shot} & \text{GNN  + FT [2]} & 66.32\small{\pm{0.80}} & 47.47\small{\pm{0.75}} & 31.61\small{\pm{0.53}} & 55.77\small{\pm{0.79}} & 35.95\small{\pm{0.58}} \newline
> & \text{GNN  + SiMT} & 67.22\small{\pm{0.79}} & 48.19\small{\pm{0.71}} & 32.47\small{\pm{0.57}} & 57.41\small{\pm{0.81}} & \mathbf{38.13\small{\pm{0.61}}} \newline
> & \text{GNN  + FT [2]+ SiMT} & \mathbf{68.02\small{\pm{0.80}}} & \mathbf{48.75\small{\pm{0.76}}} & \mathbf{32.89\small{\pm{0.69}}} & \mathbf{58.23\small{\pm{0.86}}} & 38.07\small{\pm{0.60}} \newline
> \hline
> & \text{GNN} & 80.87\small{\pm{0.56}} & 62.25\small{\pm{0.65}} & 44.28\small{\pm{0.63}} & 70.84\small{\pm{0.65}} & 52.53\small{\pm{0.59}} \newline
> \text{5-shot}& \text{GNN + FT [2]} & 81.98\small{\pm{0.55}} & 66.98\small{\pm{0.68}} & 44.90\small{\pm{0.64}} & 73.94\small{\pm{0.67}} & 53.85\small{\pm{0.62}} \newline
> & \text{GNN + SiMT } & 84.37\small{\pm{0.56} } & 68.78\small{\pm{0.69} } & 45.61\small{\pm{0.67} } & 76.73\small{\pm{0.66} } & 55.72\small{\pm{0.63} } \newline
> & \text{GNN + FT [2] + SiMT } & \mathbf{85.13\small{\pm{0.55}} } & \mathbf{70.09\small{\pm{0.67}} } & \mathbf{46.90\small{\pm{0.65}} } & \mathbf{78.15\small{\pm{0.64}} } & \mathbf{56.60\small{\pm{0.64}}} \newline
> \hline
> \end{array}
>
> [1] Flennerhag et al., Bootstrapped meta-learning, ICLR 2022\
> [2] Tseng et al., Cross-Domain Few-Shot Classification via Learned Feature-Wise Transformation, ICLR 2020\
> [3] Garcia et al., Few-shot learning with graph neural networks, ICLR 2018

---

> ### Author Response · Authors · 2022-08-02
> **Response to Reviewer siDw (1/3)**
>
> Dear reviewer siDw,
>
> We sincerely appreciate your efforts and insightful comments to improve the manuscript. We respond to each of your comments one-by-one in what follows. In the revised draft, we mark our revisions by “blue”.
>
> ------------------------------------------------
>
> **[W1] Limited novelty.**
>
> Although we agree that the momentum network is a widely used concept in the literature, our novelty is in the overall framework of SiMT, which is the first to investigate the utility of the momentum network in the meta-learning context. Moreover, as noted by Reviewer mjLb, we have novel components (e.g., dropout regularization to stabilize the self-improving process) other than the momentum network, which are demonstrated to be effective in various meta-learning applications. Furthermore, as highlighted by Reviewer x6gw, we believe that meta-learning with target models is an important, yet under-explored direction to investigate, and our efficient (yet effective) target generation technique can be an important milestone in this direction. In this regard, we think our novelty can also be found in the problem formulation itself.

---

### Official Review · Reviewer_DCr7 · 2022-07-14

**Rating:** 6
**Confidence:** 4
**Soundness:** 3 good
**Presentation:** 3 good
**Contribution:** 2 fair

**Summary:**

This paper proposes a simple algorithm to use the benefits of the knowledge distillation (KD) in meta-learning. More specifically, since it is not possible to train an individual teacher for each meta-training tasks (which could be hundreds of thousands tasks in standard setup), this method obtains a momentum network using exponential moving average of the meta-model and then adapts this momentum network into each task using support set of that task. The adapted model is used as teacher to guide the main few-shot adaptation process.


**Questions:**

Please refer to the weaknesses part.


**Limitations:**

yes

**Strengths And Weaknesses:**

Strengths:


This work introduces a very interesting idea to improve the generalization of the meta-learning algorithms. From KD point of view, it is similar to Born Again Neural Networks where a model becomes its own teacher. However, instead of training from scratch for several times, it uses momentum network to train the teacher (obtained by adaptation from this momentum network and called target network). Even though the algorithm is very simple, it works very well and provided results show improvement on baseline for different backbones and in different applications including classification, reinforcement learning and regression.

The writing is easy to follow and concepts are defined well, however, there are some concerns regarding this work.

Weaknesses:


Proposed SiMT algorithm can be considered as a regularizer which prevents the baseline meta-learning algorithm from overfitting to meta-train classes. Since the results proposed in this paper are way behind state-of-the-art (SOTA) performance, I expect to see at least the comparison with other regularizer-like methods for few-shot learning like:
- Cross-Domain Few-Shot Classification via Learned Feature-Wise Transformation, ICLR’20


In addition, as the method can be considered in the line of works that uses KD for meta-learning, comparison with previous works needs to be included:
- Towards enabling meta-learning from target models, NeurIPS’21


For few-shot classification results, as we cannot generally evaluate the whole possible set of meta-test tasks, the results usually presented as the mean and standard deviation over a batch of tasks. However, this paper just reports the std on three trials which is not accurate enough.


For RL results in Figure 2, why the std of the proposed model is much higher than the baseline? As SiMT uses momentum network, I expect to see less std in the proposed method.


For computational efficiency, the analysis is very limited and is presented just for a specific configuration.


Regarding Table 4 (Ablation studies), what does it mean to use the momentum without distillation? Do you mean that the meta-learner is updated using momentum equation?


In Fig. 3(b) Configuration is not clear. If same config, why is this different from the 3(a)?



There are also some typos in the paper:

- In line 118, it should be \mathcal{L}(\phi^{\matchal{T}_i})

- same problem with line 126, and equation (6)

- in Fig 3(a) is should be 3X faster

---

> ### Author Response · Authors · 2022-08-02
> **Response to Reviewer DCr7 (3/3)**
>
> **[W5] Computational efficiency.**
>
> We thank your constructive comments on our manuscript. To further demonstrate the computational efficiency of SiMT, we consider additional comparisons on various meta-learning methods, including ANIL, MetaSGD, and ProtoNet: we train Conv4 on the mini-ImageNet dataset under a 5-shot classification scenario. We also observed consistent efficiency gains on these setups, e.g., 30\% of training time reduced to achieve the peak accuracy of ProtoNet when using SiMT; we added the visualization of the learning curves for the efficiency comparison in revised Appendix E.4 and Fig 5. Although SiMT can be slower than the base algorithms when comparing the number of optimization steps (due to the momentum target generation), we want to note that SiMT can be more efficient in terms of time to reach the same performance.
>
> ------------------------------------------------
>
> **[W6] Clarification in Table 4: “momentum without distillation”.**
>
> The “momentum without distillation” in Table 4 refers to a scenario where there exists a momentum network updated independently from the meta-learning process by taking the exponential moving average of the meta-model (Eq 4 in the main text). The reported accuracy is the adaptation performance of this momentum network. We have clarified this in the revised draft.
>
> ------------------------------------------------
>
> **[W7] Clarification of Fig 3 (a) and (b).**
>
> We clarify that the experimental setups of Fig 3 (a) and (b) are the same, however, the figures differ in (i) the compared models and (ii) the x-axis. In Fig 3 (a), we compare the meta-model of MAML and the momentum network of MAML + SiMT, where the x-axis denotes the wall-clock time. In contrast, we compare the meta-model and momentum network of MAML + SiMT in Fig 3 (b), where the x-axis indicates the number of training steps. For a better presentation, we have revised the caption and the figures in Fig 3 by using different colors for each subfigure.
>
> ------------------------------------------------
>
> **[W8] Editorial comments**
>
> Thank you very much! We have fixed the typo in Fig 3 (a): x3 $\to$ 3x in the manuscript.
> Also, we want to clarify that we omit $\tau$ except when defining a stream of tasks for better readability (line 117, footnote 1).

---

> ### Author Response · Authors · 2022-08-02
> **Response to Reviewer DCr7 (2/3)**
>
> **[W2] Comparison with works that use KD for meta-learning [1].**
>
> We re-emphasize that our work aims to develop a method that can be used in the case where task-specific target models are not available (and thus standard $\mathcal{ST}$ protocol [1], which requires pre-trained target models, cannot be used). Moreover, even though [1] proposed an efficient way to obtain target models, the method is limited to specific domains and still requires a large amount of computations, e.g., it takes more than 6.5 GPU hours to pre-train only 10% of target models while ours spends only 2 GPU hours for the entire meta-learning process (on ProtoNet ResNet-12).
>
> Nonetheless, following the suggestion, we compare SiMT with Lu et al. [1] and train SiMT upon a pre-trained backbone ResNet-12 (on the meta-training set) following [1]. As shown in the table, SiMT consistently improves over MAML, and more intriguingly, SiMT performs even better than Lu et al. [1] when applied to MAML. We conjecture this is because SiMT can fully utilize the target model for all tasks due to its efficiency, while Lu et al. [1] should partially sample the tasks with the target model due to its computational complexity. We added this result to the revised manuscript (Appendix E.2, Table 7).
>
> \begin{array}{lccccc}
> \hline
> \text{Method}  & \text{Train cost} &\text{mini-ImageNet}  & & \text{tiered-ImageNet}  &  \newline
> & \text{(1-shot, GPU hours)} & \text{1-shot} & \text{5-shot} & \text{1-shot} &  \text{5-shot} \newline
> \hline
> \text{MAML} & 1.31 & 58.84\small{\pm{0.25}} & 74.62\small{\pm{0.38}} & 63.02\small{\pm{0.30}} & 67.26\small{\pm{0.32}} \newline
> \text{MAML + Lu et al. [1]}-5\\%~ \text{target} & 5.04 & 59.14\small{\pm{0.33}} & 75.77\small{\pm{0.29}} & 64.52\small{\pm{0.30}} & 68.39\small{\pm{0.34}} \newline
> \text{MAML + Lu et al. [1]}-10\\% ~\text{target} & 8.32 & 60.06\small{\pm{0.35}} & 76.34\small{\pm{0.42}} &  \mathbf{65.23\small{\pm{0.45}}} & 70.02\small{\pm{0.33}} \newline
> \text{MAML + SiMT} & 1.64 & \mathbf{62.05\small{\pm{0.39}}} &  \mathbf{78.77\small{\pm{0.45}}} & 63.91\small{\pm{0.32}} &  \mathbf{77.43\small{\pm{0.47}}} \newline
> \hline
> \end{array}
>
> [1] Lu et al., Towards Enabling Meta-Learning from Target Models, NeurIPS 2021
>
> ------------------------------------------------
>
> **[W3] Few-shot evaluations are usually done over a batch of tasks: three trials is not accurate enough.**
>
> We clarify that each trial indicates an average test accuracy of 2,000 meta-test tasks. Hence the reported mean and standard deviation is computed over the average test accuracies of three independently trained meta-models (in Appendix D.1 evaluation detail). Note that this evaluation setup can be found in the prior works [1,2].
>
> In the table below, we also provide the classification accuracy with 95% confidence intervals averaged over 2,000 meta-test tasks on a single (seed) meta-model where one can observe that MAML and SiMT have small variances.
>
> \begin{array}{lcccc}
> \hline
> \text{Method}  &\text{mini-ImageNet}  & & \text{tiered-ImageNet}  &  \newline
>  & \text{1-shot} & \text{5-shot} & \text{1-shot} &  \text{5-shot} \newline
> \hline
> \text{MAML} & 47.43\small{\pm{0.64}} & 63.25\small{\pm{0.56}} & 50.00\small{\pm{0.72}} & 65.89\small{\pm{0.60}} \newline
> \text{MAML + SiMT} & \mathbf{51.71\small{\pm{0.64}}} & \mathbf{68.77\small{\pm{0.53}}} & \mathbf{52.80\small{\pm{0.73}}} & \mathbf{69.57\small{\pm{0.61}}} \newline
> \hline
> \end{array}
>
> [1] Oh et al., BOIL: Towards Representation Change for Few-shot Learning, ICLR 2021\
> [2] Oswald et al., Learning where to learn: Gradient sparsity in meta and continual learning, NeurIPS 2021
>
> ------------------------------------------------
>
> **[W4] Why does SiMT show high variance in the RL experiment?**
>
> We first note that a high reward variance of the state-of-the-art RL algorithms is prevalent, being an important open problem in the field [1]. It is actually a quite common observation that the reward variance scales with the reward mean [2]; underperforming algorithms tend to have a low variance (as they consistently fail), and better performing algorithms have a high variance (as they often achieve very high performance but not always). We believe that a similar phenomenon is taking place in our experiments. Still, we emphasize that the performance gain of SiMT is significant despite its variance. We think stabilizing meta-RL performance of momentum networks is an important and interesting future direction.
>
> [1] Bjorck et al., Is High Variance Unavoidable in RL? A Case Study in Continuous Control, ICLR 2022\
> [2] Henderson et al., Deep Reinforcement Learning that Matters, AAAI 2018

---

> ### Author Response · Authors · 2022-08-02
> **Response to Reviewer DCr7 (1/3)**
>
> Dear reviewer DCr7,
>
> We sincerely appreciate your efforts and insightful comments to improve the manuscript. We respond to each of your comments one-by-one in what follows. In the revised draft, we mark our revisions by “blue”.
>
> ------------------------------------------------
>
>
> **[W1] Comparison with other regularization methods, e.g., [1].**
>
> Thank you for your suggestions. First, we would like to note that, in Appendix E.1, we provide an empirical comparison with a recent regularization method called Bootstrapped meta-learning (Bootstrap) [2]. SiMT consistently outperforms Bootstrap, as is shown in Table 5; we also give a part of the table below. We further note that SiMT has broader applicability than Bootstrap, as the latter is specialized for gradient-based meta-learning schemes, while SiMT can also be applied to other types of meta-learning, e.g., metric-based methods (such as ProtoNet).
>
> \begin{array}{lcccc}
> \hline
> \text{Method}  &\text{mini-ImageNet}  & & \text{tiered-ImageNet}  &  \newline
> & \text{1-shot} & \text{5-shot} & \text{1-shot} &  \text{5-shot} \newline
> \hline
> \text{MAML} & 47.33\small{\pm{0.45}} & 63.27\small{\pm{0.14}} & 50.19\small{\pm{0.21}} & 66.05\small{\pm{0.19}} \newline
> \text{MAML + Bootstrap [2]}  & 48.68\small{\pm{0.33}} & 68.45\small{\pm{0.40}} & 49.34\small{\pm{0.26}} & 68.84\small{\pm{0.37}} \newline
> \text{MAML + SiMT} & \mathbf{51.49\small{\pm{0.18}}} & \mathbf{68.74\small{\pm{0.12}}} & \mathbf{52.51\small{\pm{0.21}}} & \mathbf{69.58\small{\pm{0.11}}} \newline
> \hline
> \end{array}
>
> In addition, following the reviewer’s suggestion, we provide additional comparisons with the regularization method of [1], i.e., feature-wise transformation (FT) regularization (we compare with the general FT instead of the LFT variant specialized for cross-domain generalization, as the latter requires multiple datasets to learn hyperparameters). We consider the few-shot classification setup as in [1] and train ResNet-10 on the mini-ImageNet dataset, using GNN [3] as the base meta-learning method. As shown in the table below, SiMT consistently outperforms FT in all cases. More importantly, we observe that SiMT and FT can be used jointly to improve the performance further. We have added this result to the revised manuscript (in Appendix E.1, Table 6).
>
> \begin{array}{llccccc}
> \hline
> \text{Problem}  & \text{Method}  &\text{mini-ImageNet}  & \text{CUB}  &  \text{Cars}& \text{Places} & \text{Plantae}  \newline
> \hline
> & \text{GNN} & 60.77\small{\pm{0.75}} & 45.69\small{\pm{0.68}} & 31.79\small{\pm{0.51}} & 53.10\small{\pm{0.80}} & 35.60\small{\pm{0.56}} \newline
> \text{1-shot} & \text{GNN  + FT [1]} & 66.32\small{\pm{0.80}} & 47.47\small{\pm{0.75}} & 31.61\small{\pm{0.53}} & 55.77\small{\pm{0.79}} & 35.95\small{\pm{0.58}} \newline
> & \text{GNN  + SiMT} & 67.22\small{\pm{0.79}} & 48.19\small{\pm{0.71}} & 32.47\small{\pm{0.57}} & 57.41\small{\pm{0.81}} & \mathbf{38.13\small{\pm{0.61}}} \newline
> & \text{GNN  + FT [1]+ SiMT} & \mathbf{68.02\small{\pm{0.80}}} & \mathbf{48.75\small{\pm{0.76}}} & \mathbf{32.89\small{\pm{0.69}}} & \mathbf{58.23\small{\pm{0.86}}} & 38.07\small{\pm{0.60}} \newline
> \hline
> & \text{GNN} & 80.87\small{\pm{0.56}} & 62.25\small{\pm{0.65}} & 44.28\small{\pm{0.63}} & 70.84\small{\pm{0.65}} & 52.53\small{\pm{0.59}} \newline
> \text{5-shot}& \text{GNN + FT [1]} & 81.98\small{\pm{0.55}} & 66.98\small{\pm{0.68}} & 44.90\small{\pm{0.64}} & 73.94\small{\pm{0.67}} & 53.85\small{\pm{0.62}} \newline
> & \text{GNN + SiMT } & 84.37\small{\pm{0.56} } & 68.78\small{\pm{0.69} } & 45.61\small{\pm{0.67} } & 76.73\small{\pm{0.66} } & 55.72\small{\pm{0.63} } \newline
> & \text{GNN + FT [1] + SiMT } & \mathbf{85.13\small{\pm{0.55}} } & \mathbf{70.09\small{\pm{0.67}} } & \mathbf{46.90\small{\pm{0.65}} } & \mathbf{78.15\small{\pm{0.64}} } & \mathbf{56.60\small{\pm{0.64}}} \newline
> \hline
> \end{array}
>
> [1] Tseng et al., Cross-Domain Few-Shot Classification via Learned Feature-Wise Transformation, ICLR 2020\
> [2] Flennerhag et al., Bootstrapped meta-learning, ICLR 2022\
> [3] Garcia et al., Few-shot learning with graph neural networks, ICLR 2018

---

> > ### Comment · Reviewer_DCr7 · 2022-08-09
> > **Thank you for the response**
> >
> > The additional results have addressed my concerns. I have increased my rating.

---

> > > ### Author Response · Authors · 2022-08-09
> > > **Thank you for the response**
> > >
> > > Thank you for letting us know! We are happy to hear that our rebuttal addressed your concerns well.
> > >
> > > Thank you very much,\
> > > Authors

---

### Official Review · Reviewer_x6gw · 2022-07-19

**Rating:** 6
**Confidence:** 5
**Soundness:** 3 good
**Presentation:** 3 good
**Contribution:** 3 good

**Summary:**

This paper studies $\mathcal{S}$-$\mathcal{T}$ protocol in meta-learning, which means comparing a task-specific solver to a target model. The main contribution of this paper is proposing an efficient method to construct target models. The authors generate target models by adapting from the momentum network, which shows better adaptation performance than the meta-model itself. In the proposed method, the construction of target models and the training of meta-model are combined and iteratively performed. Experiment results demonstrate the effectiveness of the proposed method.

**Questions:**

Questions
1. The target models are generated according to the momentum network. Could authors evaluate the quality of target models? Do these target models perform well on their tasks?
2. Since the momentum network is updated during the meta-training process, early target models tend to be weak. Could authors give some explanations?
3. In Equation (3), the authors evaluate the task-specific solver on $\mathcal{Q}^{\tau_{i}}$, which is different from [1]. Could authors show the advantages of doing so?
4. In [1], the target models are constructed only for those hardest tasks (5% or 10%). It is interesting to see what will happen if only a part of meta-training tasks have target models. Do the number of target tasks have severe evident influences on the performance?
5. The test accuracies in Table 2 are not satisfying. For example, with ResNet-12 backbone, we often achieve 65% (or higher) accuray on 5-way 1-shot miniImageNet using ProtoNet [2,3]. Is it because the authors do not pre-train the backbone on the meta-training split?

References

[1] Lu S, Ye H J, Gan L, et al. Towards Enabling Meta-Learning from Target Models[J]. Advances in Neural Information Processing Systems, 2021, 34: 8060-8071.

[2] Ye H J, Hu H, Zhan D C, et al. Few-shot learning via embedding adaptation with set-to-set functions[C]//Proceedings of the IEEE/CVF Conference on Computer Vision and Pattern Recognition. 2020: 8808-8817.

[3] Wang Y, Chao W L, Weinberger K Q, et al. Simpleshot: Revisiting nearest-neighbor classification for few-shot learning[J]. arXiv preprint arXiv:1911.04623, 2019.

**Limitations:**

The authors have included the limitations and future work in the paper, and some other limitations are listed in the *Strengths and Weaknesses* part.

**Strengths And Weaknesses:**

Strengths
1. This paper studies an important and interesting problem in meta-learning, i.e., $\mathcal{S}$-$\mathcal{T}$ protocol, which may bypass the limitations of query sets.
2. Experiments cover both few-shot learing and reinforcement learning, which are two main applications of meta-learning.
3. This paper is well written and easy to understand.

Weaknesses
1. **Imperfect target models**. In Line 7 of Algorithm 1, we can see that the authors generate a target model according to current momentum network $\phi_{\mathrm{moment}}$. However, the momentum network is updated during the meta-training procedure, and it is weak in early epochs. The definition of target model is a strong model (expert) on a specific task, but the generated target models by the proposed algorithm are not optimal.
2. **Insufficient interpretation**. Some technical details are not discussed adequately in the paper. For example, in Equation (3), the second terms stands for the loss of $\mathcal{S}$-$\mathcal{T}$ protocol. However, the authors evaluate the task-specific solver on the query set $\mathcal{Q}^{\tau_{i}}$ rather than the support set $\mathcal{S}^{\tau_{i}}$, which is different from [1]. In Line 146, the authors claim that the predictions of target models tend to generate soft predictions, which is not verified empirically or theoretically. Apart from this, the quality of generated target models are not evaluated. Since target models are experts on corresponding tasks, it is necessary to evaluate them.

References

[1] Lu S, Ye H J, Gan L, et al. Towards Enabling Meta-Learning from Target Models[J]. Advances in Neural Information Processing Systems, 2021, 34: 8060-8071.

---

> ### Author Response · Authors · 2022-08-02
> **Response to Reviewer x6gw (2/2)**
>
> **[Q4] It is interesting to see what will happen if only a part of meta-training tasks has target models.**
>
> Thank you for your constructive suggestion. Following the suggestion, we consider generating target models for a small proportion of tasks when training SiMT. To this end, we train Conv4 on a mini-ImageNet dataset and control the ratio of tasks with the target model. As shown in the table, SiMT shows consistent improvement even with a small portion of task models, e.g., 5% of target models show 1% improvement in 1-shot adaptation. However, remark that one can easily generate target models for all tasks with SiMT, hence, one does not need to use few targets in practice. We included this result in the revised manuscript (Table 8 in Appendix E.3).
>
> \begin{array}{lcccccc}
> \hline
> \text{Proportion of tasks with targets}  & 0\\% & 5\\%  & 10\\% & 25\\%  & 50\\% & 100\\% \newline
> \hline
> \text{1-shot} & 48.98\small{\pm0.32} & 49.83\small{\pm0.12} & 49.92\small{\pm0.23}& 50.42\small{\pm0.67} & 50.55\small{\pm0.41} & \mathbf{51.49\small{\pm0.18}} \newline
> \text{5-shot} &66.12\small{\pm{0.21}} & 66.81\small{\pm{0.26}} &66.99\small{\pm{0.24}} &67.15\small{\pm{0.18}}& 67.49\small{\pm{0.39}}  & \mathbf{68.74\small{\pm0.12}}\newline
> \hline
> \end{array}
>
> ------------------------------------------------
>
> **[Q5] The test accuracies are low compared to the prior works that use a pre-trained backbone.**
>
> Thank you for pointing this out! We did not use a pre-trained backbone (on meta-training split) when training SiMT, so the results can be low compared to the prior works trained on a pre-trained network. To verify that SiMT is also effective even with the pre-trained backbone, we additionally run the following experiment: train SiMT over MAML with the pre-trained ResNet-12 backbone (provided in [1]). As shown in the table below, SiMT also consistently improves the performance over MAML in this setup. More intriguingly, SiMT performs even better than Lu et al. [2] (i.e., training with pre-trained target models) in some cases. We conjecture this is because SiMT can fully utilize the target model for all tasks due to its efficiency, while Lu et al. [2] should partially sample the tasks with the target model due to its computational complexity. We added this result in the revised manuscript (Table 7 in Appendix E.2).
>
> \begin{array}{lccccc}
> \hline
> \text{Method}  & \text{Train cost} &\text{mini-ImageNet}  & & \text{tiered-ImageNet}  &  \newline
> & \text{(1-shot, GPU hours)} & \text{1-shot} & \text{5-shot} & \text{1-shot} &  \text{5-shot} \newline
> \hline
> \text{MAML} & 1.31 & 58.84\small{\pm{0.25}} & 74.62\small{\pm{0.38}} & 63.02\small{\pm{0.30}} & 67.26\small{\pm{0.32}} \newline
> \text{MAML + Lu et al. [2]}-5\\%~ \text{target} & 5.04 & 59.14\small{\pm{0.33}} & 75.77\small{\pm{0.29}} & 64.52\small{\pm{0.30}} & 68.39\small{\pm{0.34}} \newline
> \text{MAML + Lu et al. [2]}-10\\% ~\text{target} & 8.32 & 60.06\small{\pm{0.35}} & 76.34\small{\pm{0.42}} &  \mathbf{65.23\small{\pm{0.45}}} & 70.02\small{\pm{0.33}} \newline
> \text{MAML + SiMT} & 1.64 & \mathbf{62.05\small{\pm{0.39}}} &  \mathbf{78.77\small{\pm{0.45}}} & 63.91\small{\pm{0.32}} &  \mathbf{77.43\small{\pm{0.47}}} \newline
> \hline
> \end{array}
>
> [1] Ye et al., Few-Shot Learning via Embedding Adaptation with Set-to-Set Functions, CVPR 2020\
> [2] Lu et al., Towards Enabling Meta-Learning from Target Models, NeurIPS 2021

---

> ### Author Response · Authors · 2022-08-02
> **Response to Reviewer x6gw (1/2)**
>
> Dear reviewer x6gw,
>
> We sincerely appreciate your efforts and insightful comments to improve the manuscript. We respond to each of your comments one-by-one in what follows. In the revised draft, we mark our revisions by “blue”.
>
> ----------------
> **[W1 & Q2] The proposed target model can be weak in the early training process.**
>
> It is true that the performance of target models used in SiMT may not match the performance of target task experts that are trained exclusively for that task. The goal of our paper, however, is to provide a method which can be used whenever such target task experts are not available at our hand; training such models requires a lot of computation, or even impossible for RL tasks. In this respect, the suboptimality of momentum targets should be considered a constraint imposed by the problem setup, rather than the suboptimality of our design.
>
> In addition, one thing we would like to note is that it is still questionable whether the high performance of target models is essential for the good performance of students [1]. In fact, a recent observation from the self-supervised literature is that what is important is the teacher being “consistently better” than the student, rather than its absolute performance [2]. This property holds true for the momentum target of SiMT, as we observed in Fig 3 (b).
>
> [1] Cho et al., On the Efficacy of Knowledge Distillation, CVPR 2019\
> [2] Caron et al., Emerging Properties in Self-Supervised Vision Transformers, ICCV 2021
> ----------------
>
> **[W2 & Q3] Insufficient discussions on using query sets for S-T protocol loss.**
>
> Thank you for pointing this out. We have enhanced the manuscript with more discussions on this matter. The added discussions can be summarized as follows: We use the distillation loss evaluated on the query set (as opposed to the support set, as in [1]) to improve the generalization performance of the student model. In particular, the predictions of adapted models on query set samples are softer (i.e., having less confidence) than on support set samples, and such soft predictions are known to be beneficial on the generalization performance of the student model in the knowledge distillation literature [2,3]. At the early stage of our research, we also tried using the support set for the distillation, but, we found that the query set was better (as shown in the table below where we train Conv4 on the mini-ImageNet dataset).
>
> \begin{array}{lcc}
> \hline
> \text{Distillation}  & \text{1-shot} & \text{5-shot}  \newline
> \hline
> \text{Support set} ~\mathcal{S} & 50.98\small{\pm0.12} & 67.56\small{\pm0.15} \newline
> \text{Query set} ~\mathcal{Q} & \mathbf{51.49\small{\pm0.18}} & \mathbf{68.74\small{\pm0.12}}\newline
> \hline
> \end{array}
>
> [1] Lu et al., Towards Enabling Meta-Learning from Target Models, NeurIPS 2021\
> [2] Yuen et al., Revisiting knowledge distillation via label smoothing regularization, CVPR 2020\
> [3] Tang et al., Understanding and improving knowledge distillation, arXiv 2020
>
> ----------------
>
> **[W2 & Q1] The quality of generated target models are not evaluated.**
>
> We first remark that the quality of the generated target model is given in Fig 3 (b) in the submitted draft, i.e., we measured the (meta-validation) adaptation performance of the momentum network while training SiMT, where it consistently shows a better adaptation performance than the meta-model.
>
>
> To further analyze the quality of the target model, we compare the momentum target with pre-trained target models in [1]: for SiMT, we train ResNet-12 on 5-shot mini-ImageNet and report the last checkpoint's adaptation performance on 1,000 meta-train tasks. Here, (somewhat obviously) the pre-trained target model shows better performance than the momentum target: the pre-trained target model shows 99.37\% whereas the momentum target shows 82.30\%. Clearly, it would be nice to have a pre-trained target for all tasks but note that a momentum target is an efficient substitute as it still consistently shows better performance than the student model, which is known to be important in improving the generalization of the student model [2].
>
> [1] Lu et al., Towards Enabling Meta-Learning from Target Models, NeurIPS 2021\
> [2] Caron et al., Emerging Properties in Self-Supervised Vision Transformers, ICCV 2021

---

### Author Response · Authors · 2022-08-02
**General Response**

Dear reviewers and AC,

We sincerely appreciate your valuable time and effort spent reviewing our manuscript.

As reviewers highlighted, we believe our paper tackles an interesting and important problem (x6gw), and provides a novel (mjLB) and effective (DCr7, siDw, mjLB) framework for meta-learning, validated with extensive evaluations (all reviewers) followed by a clear presentation (x6gw, DCr7, mjLB).

We appreciate your constructive comments on our manuscript. In response to the comments, we have carefully revised and enhanced the manuscript with the following additional discussions and experiments:
- Provide an overview of terminologies used in the paper (Appendix A), and a clearer description of the momentum network (Section 1)
- Clearer description of using a query set for the distillation (Section 4)
- Detailed description of experimental details in the captions (Table 4, Fig 3)
- Additional comparison with a recent regularization scheme, called FT (Appendix E.1, Table 6)
- Comparison with pre-trained target models (Appendix E.2, Table 7)
- Experiments using a small proportion of tasks with momentum targets in SiMT (Appendix E.3, Table 8)
- Additional computational efficiency analysis (Appendix E.4, Fig 5)

These updates are temporarily highlighted in “blue” for your convenience to check.

We sincerely believe that SiMT can be a useful addition to the NeurIPS community, in particular, due to the above revision helping us better deliver the effectiveness of our method.

Thank you very much !\
Authors.

---

### Author Response · Authors · 2022-08-07
**A gentle reminder**

Dear reviewers,

Thank you for your time and efforts again in reviewing our paper.

We kindly remind that the discussion period will end soon (in a few days).

We believe that we sincerely and successfully address your comments, with the results of the supporting experiments.

If you have any further concerns or questions, please do not hesitate to let us know.

Thank you very much!

Authors

---

### Meta-Review · Area_Chair_LMAy · 2022-08-23

**Recommendation:** Accept
**Confidence:** Certain

**Metareview:**

This submission proposes a strategy to improve meta-learning that can be applied to many different base meta-learning methods. The base meta-learning method is used to independently adapt both the online network whose parameters are being optimized and a momentum network constructed by taking an exponential moving average of the online network's weights, and a distillation loss is used to encourage the adapted online network (with dropout on its parameters) to match the adapted momentum network. Extensive experiments demonstrate that the proposed method improves performance of several base meta-learning methods, and that each component of the method is necessary to attain optimal performance. Reviewers initially praised the idea and empirical evaluation, but noted that the results obtained were far from state-of-the-art, and asked for additional experiments with better-performing base meta-learning methods. The authors provided these experiments during the response period, and the reviewers now unanimously recommend acceptance. The AC agrees with the reviewers' assessment.

**Award:**

No

---

### Decision · Program_Chairs · 2022-09-14

Accept